# Self-Exploring Language Models:
# Active Preference Elicitation for Online Alignment

**Shenao Zhang[1], Donghan Yu[2], Hiteshi Sharma[2], Han Zhong[3], Zhihan Liu[1]**
**Ziyi Yang[2], Shuohang Wang[2], Hany Hassan[2], Zhaoran Wang[1]**
[1]*Northwestern University*    [2]*Microsoft*    [3]*Peking University*

**Reviewed on OpenReview:** *https://openreview.net/forum?id=FoQK84nwY3*

## Abstract

Preference optimization, particularly through Reinforcement Learning from Human Feedback (RLHF), has achieved significant success in aligning Large Language Models (LLMs) to adhere to human intentions. Unlike offline alignment with a fixed dataset, online feedback collection from humans or AI on model generations typically leads to more capable reward models and better-aligned LLMs through an iterative process. However, achieving a globally accurate reward model requires systematic exploration to generate diverse responses that span the vast space of natural language. Random sampling from standard reward-maximizing LLMs alone is insufficient to fulfill this requirement. To address this issue, we propose a bilevel objective optimistically biased towards potentially high-reward responses to actively explore out-of-distribution regions. By solving the inner-level problem with the reparameterized reward function, the resulting algorithm, named *Self-Exploring Language Models* (SELM), eliminates the need for a separate RM and iteratively updates the LLM with a straightforward objective. Compared to *Direct Preference Optimization* (DPO), the SELM objective reduces indiscriminate favor of unseen extrapolations and enhances exploration efficiency. Our experimental results demonstrate that when fine-tuned on Zephyr-7B-SFT and Llama-3-8B-Instruct models, SELM significantly boosts the performance on instruction-following benchmarks such as MT-Bench and AlpacaEval 2.0, as well as various standard academic benchmarks in different settings.

## 1 Introduction

Large Language Models (LLMs) have recently achieved significant success largely due to their ability to follow instructions with human intent. As the de facto method for aligning LLMs, Reinforcement Learning from Human Feedback (RLHF) works by maximizing the reward function, either a separate model (Ouyang et al., 2022; Bai et al., 2022; Gao et al., 2023) or reparameterized by the LLM policy (Rafailov et al., 2024b;a; Azar et al., 2023; Zhao et al., 2023), which is learned from the prompt-response preference data labeled by humans. The key to the success of alignment is the response *diversity* within the preference data, which prevents reward models (RMs) from getting stuck in local optima, thereby producing more capable language models.

Offline alignment methods (Rafailov et al., 2024b; Tang et al., 2024) attempt to manually construct diverse responses for fixed prompts (Cui et al., 2023; Ivison et al., 2023; Zhu et al., 2023), which, unfortunately, struggles to span the nearly infinite space of natural language. On the other hand, online alignment follows an *iterative* procedure: sampling responses from the LLM and receiving feedback to form new preference data for RM training (Ouyang et al., 2022; Guo et al., 2024). The former step helps explore out-of-distribution (OOD) regions through randomness in sampling. However, in standard online RLHF frameworks, maximizing

the expected reward learned from the collected data is the only objective for the LLM, sampling from which often leads to responses clustered around local optima. This passive exploration mechanism can suffer from overfitting and premature convergence, leaving the potentially high-reward regions unexplored.

To address this issue, we propose an active exploration method for online alignment that elicits novel favorable responses. In its simplest form, an optimism term $\alpha \max_y r(x, y)$ is added to the reward-fitting loss (e.g., the negative log-likelihood $\mathcal{L}_{\text{lr}}$ on dataset $\mathcal{D}$), resulting in a bilevel optimization objective for the *reward* model $r$:

$$\max_r \max_y \alpha r(x, y) - \mathcal{L}_{\text{lr}}(r; \mathcal{D}), \tag{1}$$

where $\alpha$ is a hyperparameter controlling the degree of optimism. The intuition is illustrated in Figure 1. Specifically, minimizing the vanilla reward-fitting loss $\mathcal{L}_{\text{lr}}$ is likely to give a locally accurate RM that overfits the observed data and gets stuck in local minima. Random sampling from this vanilla RM may take a long time to explore the OOD regions that contain the best response. By incorporating the optimism term, we obtain an RM that *both* fits the data well and has a large $\max_y r(x, y)$. This ensures that the greedy response $y_u$ from it is either globally optimal when uncertainty in high-reward regions is eliminated, or potentially good in unexplored areas where $r(x, y_u)$ can be arbitrarily huge due to the relaxed reward-fitting loss. Feedback from humans on these responses $y_u$ can then reduce uncertainty and train a more accurate RM.

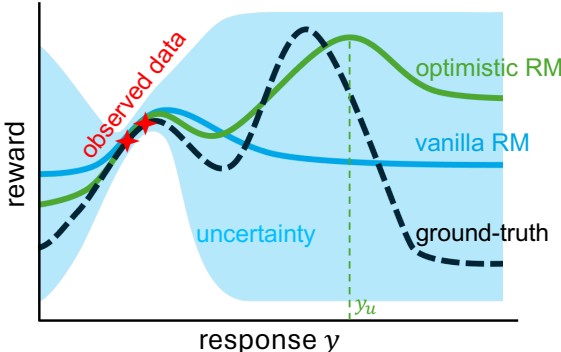

Figure 1: Intuition of our method. For a fixed prompt $x$, a reward model $r(x, y)$ tries to fit the ground-truth reward $r^*(x, y)$. The blue and green RMs are equally good when using standard reward-fitting loss $\mathcal{L}_{\text{lr}}$, since the observed preference data (red stars) are fitted equally well. However, the green RM has a larger $\max_y r(x, y)$ and thus a lower optimistically biased loss $\mathcal{L}_{\text{lr}} - \alpha \max_y r(x, y)$. Therefore, the response $y_u$ that maximizes the optimistic RM can be elicited and proceeded for human feedback to reduce the uncertainty.

In this paper, we formulate this idea within the context of online *direct* alignment, where the LLM is iteratively updated without a separate RM. We first introduce two modifications to the bilevel RM objective in (1), namely adding KL constraints and using relative maximum reward. Then we derive a simple LLM training objective by applying the closed-form solution of the inner-level problem and reparameterizing the reward with the LLM policy. The resulting iterative algorithm is called *Self-Exploring Language Models* (SELM). We show that the policy gradient of SELM is biased towards more rewarding areas. Furthermore, by reducing the chance of generating responses that are assigned low implicit rewards, SELM mitigates the *indiscriminate* favoring of unseen extrapolations in DPO (Rafailov et al., 2024b;a) and enhances exploration efficiency.

In experiments, we implement SELM using Zephyr-7B-SFT (Tunstall et al., 2023b) and Llama-3-8B-Instruct (Meta, 2024) as base models. By fine-tuning solely on the UltraFeedback (Cui et al., 2023) dataset and using the small-sized PairRM (Jiang et al., 2023) for iterative AI feedback, SELM boosts the performance of Zephyr-7B-SFT and Llama-3-8B-Instruct by a large margin on AlpacaEval 2.0 (Dubois et al., 2024) (+16.24% and +11.75% LC win rates) and MT-Bench (Zheng et al., 2024) (+2.31 and +0.32) in 3 iterations. SELM also demonstrates strong performance on standard academic benchmarks and achieves higher pairwise LC win rates against the strong iterative DPO baseline.

## 2   Related Work

**Data Synthesis for LLMs.**   A key challenge for fine-tuning language models to align with users' intentions lies in the collection of demonstrations, including both the instruction-following expert data and the preference data. Gathering such data from human labelers is expensive and sometimes suffers from variant quality (Ouyang et al., 2022; Köpf et al., 2024). To address this issue, synthetic data (Liu et al., 2024a) has been used for aligning LLMs. One line of work focuses on generating plausible instruction prompts for unlabeled data by regarding the target output as instruction-following responses (Li et al., 2023a; Wu et al., 2023; Josifoski et al., 2023; Taori et al., 2023; Li et al., 2024a). Besides, high-quality data can also be distilled from strong models for fine-tuning weaker ones (Gunasekar et al., 2023; Abdin et al., 2024; Li et al., 2023b; Ding et al., 2023; Peng et al., 2023). To construct synthetic datasets for offline RLHF, a popular pipeline (Cui et al., 2023; Tunstall et al., 2023b; Wang et al., 2024b; Ivison et al., 2023; Zhu et al., 2023) involves selecting responses sampled from *various* LLMs on a set of prompts in the hope to increase the diversity of the data that can span the whole language space. However, data manually collected in such a passive way does not consider what improves the model most through its training, leaving the potentially high-reward regions unexplored.

**Iterative Online Preference Optimization.**   Compared to offline RLHF algorithms (Rafailov et al., 2024b; Zhao et al., 2023; Azar et al., 2023) that collect preference datasets ahead of training, online RLHF (Ouyang et al., 2022; Guo et al., 2024), especially the iterative/batched online RLHF (Bai et al., 2022; Xu et al., 2023; Chen et al., 2022; Gulcehre et al., 2023; Hoang Tran, 2024; Xiong et al., 2023; Calandriello et al., 2024; Rosset et al., 2024) has the potential to gather better and better synthetic data as the model improves. As a special case, self-aligned models match their responses with desired behaviors, such as model-generated feedback (Yuan et al., 2024; Yuanzhe Pang et al., 2024; Sun et al., 2024; Wang et al., 2024a). Unfortunately, the above methods still passively explore by relying on the randomness during sampling and easily get stuck at local optima and overfit to the current data due to the vast space of natural language. A notable exception is Dwaracherla et al. (2024), which proposed to use ensembles of RMs to approximately measure the uncertainty for posterior-sampling active exploration. On the contrary, our method explores based on the optimistic bias and does not estimate the uncertainty explicitly, bypassing the need to fit multiple RMs.

**Active Exploration.**   In fact, active exploration has been widely studied beyond LLMs. Similar to Dwaracherla et al. (2024), most existing sample-efficient RL algorithms first estimate the uncertainty of the environment using historical data and then either plan with optimism (Auer, 2002; Russo & Van Roy, 2013; Jin et al., 2020; Mehta et al., 2023; Das et al., 2024), or select the optimal action from a statistically plausibly set of values sampled from the posterior distribution (Strens, 2000; Osband et al., 2013; 2023; Zhang, 2022; Li et al., 2024c). The proposed self-exploration objective can be categorized as an optimism-based exploration method. However, most previous works involve explicitly maintaining confidence sets that contain the ground truth with high probability, which poses considerable challenges for implementation especially in high-dimensional spaces such as natural language. In contrast, our method only optimizes a single objective that integrates reward estimation and planning to automatically balance exploration and exploitation.. Ensemble methods (Osband et al., 2024; Chua et al., 2018; Lu & Van Roy, 2017) can serve as approximations to estimate the uncertainty but are still computationally inefficient.

**Concurrent Work.**   We highlight the concurrent work (to the first version of the current paper) of Xie et al. (2024); Cen et al. (2024); Liu et al. (2024c), among which Xie et al. (2024) establishes the first analysis of the sample complexity of a DPO algorithm in the online setting of RLHF (formulated as MDPs). All of them focus on incorporating an SFT loss or a similar term (as bonus or penalty) alongside the DPO loss as an optimistic or pessimistic adjustment in the online or offline setting, respectively. Xie et al. (2024); Cen et al. (2024) and the current paper focus on the former, while Liu et al. (2024c) focuses on the latter. In the second version of the current paper, we provide the sample complexity of SELM following the proof technique of Xie et al. (2024). Through a reduction technique from Xie et al. (2024), we show how to connect the sample complexity of SELM to that of existing RL algorithms (Zhong et al., 2022; Liu et al., 2024b), which are not tailored to RLHF but enjoy strong theoretical guarantees.

## 3 Background

**Large Language Models.** A language model $\pi \in \Delta_{\mathcal{Y}}^{\mathcal{X}}$ typically takes the prompt $x \in \mathcal{X}$ as input and outputs the response $y \in \mathcal{Y}$. Here, $\mathcal{X}$ and $\mathcal{Y}$ are finite spaces of prompts and responses, respectively. Given the prompt $x \in \mathcal{X}$, a discrete probability distribution $\pi(\cdot \mid x) \in \Delta_{\mathcal{Y}}$ is generated, where $\Delta_{\mathcal{Y}}$ is the set of discrete distributions over $\mathcal{Y}$. After pretraining and Supervised Fine-Tuning (SFT), preference alignment is employed to enhance the ability of the language model to follow instructions with human intentions.

**Reinforcement Learning from Human Feedback (RLHF).** Standard RLHF frameworks consist of learning a reward model and then optimizing the LLM policy using the learned reward.

Specifically, a point-wise reward $r(x, y) : \mathcal{X} \times \mathcal{Y} \to \mathcal{R}$ represents the Elo score (Elo & Sloan, 1978) of the response $y$ given the prompt $x$. Then the preference distribution can be expressed by the Bradley-Terry model that distinguishes between the preferred response $y_w$ and the dispreferred response $y_l$ given prompt $x$, denoted as $y_w \succ y_l \mid x$, using the logistic function $\sigma$:

$$
\begin{aligned}
p(y_w \succ y_l \mid x) &:= \mathbb{E}_h\big[\mathbb{1}(h \text{ prefers } y_w \text{ over } y_l \text{ given } x)\big] \\
&= \sigma\big(r(x, y_w) - r(x, y_l)\big) = \frac{\exp\big(r(x, y_w)\big)}{\exp\big(r(x, y_w)\big) + \exp\big(r(x, y_l)\big)},
\end{aligned}
\tag{2}
$$

where $h$ denotes the human rater and the expectation is over $h$ to account for the randomness of the choices of human raters we ask for their preference. When provided a static dataset of $N$ comparisons $\mathcal{D} = \{x_i, y_{w,i}, y_{l,i}\}_{i=1}^N$, the parameterized reward model can be learned by minimizing the following negative log-likelihood loss:

$$
\mathcal{L}_{\text{lr}}(r; \mathcal{D}) = -\mathbb{E}_{(x, y_w, y_l) \sim \mathcal{D}}\big[\log \sigma\big(r(x, y_w) - r(x, y_l)\big)\big].
\tag{3}
$$

In the following sections, we mainly consider the above formula of reward models that is learned from preference pairs. Using the learned reward, the LLM policy $\pi \in \Delta_{\mathcal{Y}}^{\mathcal{X}}$ is optimized with reinforcement learning (RL) to maximize the expected reward while maintaining a small deviation from some base reference policy $\pi_{\text{ref}}$, i.e., maximizing the following objective

$$
\mathcal{J}(\pi) = \mathbb{E}_{x \sim \mathcal{D}, y \sim \pi(\cdot \mid x)}\big[r(x, y)\big] - \beta \mathbb{D}_{\text{KL}}(\pi \,\|\, \pi_{\text{ref}}),
\tag{4}
$$

where $\beta$ is a hyperparameter and $\mathbb{D}_{\text{KL}}(\pi \,\|\, \pi_{\text{ref}}) := \mathbb{E}_{x \sim \mathcal{D}}[\text{KL}(\pi(\cdot \mid x) \,\|\, \pi_{\text{ref}}(\cdot \mid x))]$ is the expected Kullback-Leibler (KL) divergence. An ideal $\pi_{\text{ref}}$ is the policy that helps mitigate the distribution shift issue (Rafailov et al., 2024b; Guo et al., 2024) between the true preference distribution and the policy $\pi$ during the off-policy RL training. Since we only have access to the dataset $\mathcal{D}$ sampled from the unavailable true preference distribution, $\pi_{\text{ref}}$ can be obtained by fine-tuning on the preferred responses in $\mathcal{D}$ or simply setting $\pi_{\text{ref}} = \pi^{\text{SFT}}$ and performing RLHF based on the SFT model.

**Direct Alignment from Preference.** With the motivation to get rid of a separate reward model, which is computationally costly to train, recent works (Rafailov et al., 2024b; Azar et al., 2023; Zhao et al., 2023; Tunstall et al., 2023b; Ethayarajh et al., 2024) derived the preference loss as a function of the policy by changing variables. Among them, DPO (Rafailov et al., 2024b) shows that when the BT model in (2) can perfectly fit the preference, the RLHF objective in (4) is equivalent to:

$$
\mathcal{L}_{\text{DPO}}(\pi; \mathcal{D}) = -\mathbb{E}_{(x, y_w, y_l) \sim \mathcal{D}}\left[\log \sigma\left(\beta \log \frac{\pi(y_w \mid x)}{\pi_{\text{ref}}(y_w \mid x)} - \beta \log \frac{\pi(y_l \mid x)}{\pi_{\text{ref}}(y_l \mid x)}\right)\right].
$$

# 4 Self-Exploring Language Models

## 4.1 RM-Free Objective for Active Exploration

In this section, we present several modifications to the optimistically biased objective (1) motivated in the introduction. Then we derive an RM-free objective for the LLM policy and analyze how active exploration works by examining its gradient.

First, we consider the equivalence of (1): $\max_r -\mathcal{L}_{\mathrm{lr}}(r; \mathcal{D}) + \alpha \max_\pi \mathbb{E}_{y \sim \pi}[r(x, y)]$, where the inner $\pi$ is deterministic when optimal. To account for the change of $\pi$ relative to the reference policy $\pi_{\mathrm{ref}}$, we introduce two modifications: (1) replacing the optimistic bias term $\max_\pi \mathbb{E}_{y \sim \pi}[r(x, y)]$ with $\max_\pi \mathbb{E}_{y \sim \pi, y' \sim \pi_{\mathrm{ref}}}[r(x, y) - r(x, y')]$, and (2) incorporating a KL-divergence loss term between $\pi$ and $\pi_{\mathrm{ref}}$ to minimize the deviation between $\pi$ and $\pi_{\mathrm{ref}}$. Switching to relative rewards creates a stronger incentive for exploration, as they become significant only when $\pi$ discovers responses that outperform those from $\pi_{\mathrm{ref}}$. Intuitively, it diverges from (1) primarily in scenarios where the reward of responses from $\pi$ is high, but the relative reward is low. This signals that responses from $\pi_{\mathrm{ref}}$ also receive high rewards and further exploration in this region is unnecessary.

Formally, for the reward $r$, the bilevel optimization problem with optimism is formulated as:

$$\max_r -\mathcal{L}_{\mathrm{lr}}(r; \mathcal{D}_t) + \alpha \max_\pi \left( \underbrace{\mathbb{E}_{\substack{x \sim \mathcal{D}_t, y \sim \pi(\cdot|x) \\ y' \sim \pi_{\mathrm{ref}}(\cdot|x)}} \Big[ r(x, y) - r(x, y') \Big] - \beta \mathbb{D}_{\mathrm{KL}}(\pi \,\|\, \pi_{\mathrm{ref}})}_{\mathcal{F}(\pi; r)} \right), \tag{5}$$

where $\mathcal{D}_t = \{x_i, y_{w,i}^t, y_{l,i}^t\}_{i=1}^N$ is the associated dataset at iteration $t$ and $\mathcal{L}_{\mathrm{lr}}$ is defined in (3). The nested optimization in (5) can be handled by first solving the inner optimization $\mathcal{F}(\pi; r)$ to obtain $\pi_r$ that is optimal under $r$. The solution is as follows and we defer all the derivations in this section to Appendix A.

$$\pi_r(y \mid x) := \operatorname*{argmax}_\pi \mathcal{F}(\pi; r) = \frac{1}{Z_r(x)} \pi_{\mathrm{ref}}(y \mid x) \exp\big(r(x, y)/\beta\big),$$

where the partition function $Z_r(x) = \sum_y \pi_{\mathrm{ref}}(y|x) \exp(r(x, y)/\beta)$. By substituting $\pi = \pi_r$ into $\mathcal{F}(\pi; r)$, we can rewrite the bilevel objective in (5) as a single-level one:

$$\max_r -\mathcal{L}_{\mathrm{lr}}(r; \mathcal{D}_t) + \alpha \mathcal{F}(\pi_r; r).$$

Following the implicit reward formulation in DPO, we reparameterize the reward function with $\theta \in \Theta$ as $\widehat{r}_\theta(x, y) = \beta(\log \pi_\theta(y \mid x) - \log \pi_{\mathrm{ref}}(y \mid x))$, which is the optimal solution of (4) and can express *all* reward classes consistent with the BT model as proved in (Rafailov et al., 2024b). With the above change of variable, we obtain the RM-free objective for direct preference alignment with optimism:

$$\max_{\pi_\theta} -\mathcal{L}_{\mathrm{DPO}}(\pi_\theta; \mathcal{D}_t) - \alpha\beta \mathbb{E}_{x \sim \mathcal{D}, y \sim \pi_{\mathrm{ref}}(\cdot|x)} \big[ \log \pi_\theta(y \mid x) \big]. \tag{6}$$

We now analyze how this new objective encourages active exploration. Specifically, we derive the gradient of (6) with respect to $\theta$ as

$$\underbrace{\beta \mathbb{E}_{(x, y_w, y_l) \sim \mathcal{D}_t} \Big[ \sigma\big(\widehat{r}_\theta(x, y_l) - \widehat{r}_\theta(x, y_w)\big) \big(\nabla_\theta \log \pi_\theta(y_w \mid x) - \nabla_\theta \log \pi_\theta(y_l \mid x)\big) \Big]}_{-\nabla_\theta \mathcal{L}_{\mathrm{DPO}}(\pi_\theta; \mathcal{D}_t)}$$
$$- \alpha\beta \mathbb{E}_{x \sim \mathcal{D}, y \sim \pi_\theta(\cdot|x)} \big[ \exp\big(-\widehat{r}_\theta(x, y)/\beta\big) \nabla_\theta \log \pi_\theta(y \mid x) \big]. \tag{7}$$

We note that the second line, corresponding to the gradient of the optimism term, decreases the log-likelihood of response $y$ generated by $\pi_\theta$ that has a high value of $\exp(-\widehat{r}_\theta(x, y)/\beta)$. Therefore, the added optimism term biases the gradient toward parameter regions that can elicit responses $y$ with high implicit reward $\widehat{r}_\theta$, consistent with our intuition outlined in Figure 1.

This also explains why $\mathbb{E}_{\pi_{\text{ref}}}[\log \pi_\theta]$ is minimized in our objective (6), which is equivalent to maximizing the KL divergence between $\pi_{\text{ref}}$ and $\pi_\theta$, while the reverse KL in the policy optimization objective (4) is minimized. For the DPO gradient $\nabla_\theta \mathcal{L}_{\text{DPO}}(\pi_\theta; \mathcal{D}_t)$, the degree of deviation of policy $\pi_\theta$ from $\pi_{\text{ref}}$ only affects the preference estimated with $\widehat{r}_\theta$. In other words, $\sigma(\widehat{r}_\theta(x, y_l) - \widehat{r}_\theta(x, y_w))$ is a scalar value and the policy deviation only determines the *step size* of the policy gradient, instead of its *direction*. On the other hand, our added exploration term directly controls the direction of the gradient toward potentially more rewarding areas while still fitting the preference data in $\mathcal{D}_t$. As more feedback data is collected iteratively, deviating from the unbiasedly fitted model incurs a higher DPO loss, which ultimately dominates our objective at convergence. This mechanism ensures that the resulting LLM effectively balances between exploring novel responses and exploiting previously observed ones, leading to a more accurate and aligned model.

## 4.2 Algorithm

With the optimistically biased objective derived above, the language model can actively generate OOD responses worth exploring. Human or AI feedback follows to reduce the uncertainty in these regions. These two steps are executed iteratively to get a more and more aligned model.

In practice, we split the offline preference dataset into three portions with equal sizes, one for each iteration. Besides, we use AI rankers, such as external RMs, to provide feedback on the model-generated response and the original chosen, rejected responses. The complete pseudocode of our algorithm, named *Self-Exploring Language Models* (SELM), is outlined in Algorithm 1. In experiments, we set $T$ to 3.

---

**Algorithm 1** Self-Exploring Language Models (SELM)

---

**Input:** Reference model $\pi_{\text{ref}}$, preference dataset $\mathcal{D}$, online iterations $T$, optimism coefficient $\alpha$.
1: **for** iteration $t = 1, 2, \ldots, T$ **do**
2:     Set $\mathcal{D}_t$ as the $t$-th portion of $\mathcal{D}$ and generate $y \sim \pi_{\text{ref}}(\cdot \mid x)$ for each prompt $x$ in $\mathcal{D}_t$.
3:     Rank $\{y, y_w, y_l\}$ and update $\mathcal{D}_t$ to contain the best (chosen) and worst (rejected) responses.
4:     Train the LLM $\pi_{\theta_t} = \text{argmax}_{\pi_\theta}\{-\mathcal{L}_{\text{DPO}}(\pi_\theta; \mathcal{D}_t) - \alpha\mathbb{E}_{x \sim \mathcal{D}_t}[\log \pi_\theta(y \mid x)]\}$, let $\pi_{\text{ref}} = \pi_{\theta_t}$.
5: **end for**

---

# 5 Analysis

## 5.1 Self-Exploration Reduces Indiscriminate Favor of Unseen Extrapolations

It has been observed recently (Rafailov et al., 2024a; Pal et al., 2024; Xu et al., 2024) that DPO decreases the likelihood of responses generated by the reference policy. It is because for any prompt $x$, at convergence when $\pi_\theta \neq \pi_{\text{ref}}$, it holds that

$$\mathbb{E}_{y \sim \pi_{\text{ref}}}[\widehat{r}_\theta(x, y)/\beta] = \mathbb{E}_{y \sim \pi_{\text{ref}}}[\log \pi_\theta(y \mid x) - \log \pi_{\text{ref}}(y \mid x)] = -\text{KL}(\pi_{\text{ref}}(\cdot \mid x) \| \pi_\theta(\cdot \mid x)) < 0,$$

while at the beginning of training when $\pi_\theta = \pi_{\text{ref}}$, the above terms are zero. Thus, the expected implicit reward $\widehat{r}_\theta$ as well as the likelihood of $\pi_\theta$ will decrease on the reference model's responses. This indicates that DPO stimulates a biased distribution favoring unseen extrapolated responses. In the online iterative setting that we consider, the LLM policy generates responses and receives preference feedback alternately, where biasing toward OOD regions may sometimes help discover outstanding novel responses. However, DPO *indiscriminately* favors unseen extrapolations and *passively* explores based purely on the randomness inherent in sampling from the LLM. As a consequence, the vast space of natural language makes it almost impossible to exhaustively explore all the possible responses and identify those that most effectively benefit alignment.

Next, we demonstrate that SELM mitigates this issue by performing guided exploration. Specifically, consider the proposed self-exploration objective in (6) that is derived from (5), which, in addition to the standard

DPO loss, also minimizes $\mathbb{E}_{x,y\sim\pi_{\text{ref}}}[\log \pi_\theta(y \mid x)]$. We now investigate how the probability distribution changes with this term incorporated.

**Theorem 5.1.** For any $\rho \in \Theta$ in the policy parameter space, let $\widehat{r}_\rho(x,y) = \beta(\log \pi_\rho(y \mid x) - \log \pi_{\text{ref}}(y \mid x))$ be the reparameterized implicit reward. Denote $\pi_\rho^{\min}$ as the policy that minimizes the expected implicit reward under the KL constraint, i.e.,

$$\pi_\rho^{\min}(\cdot \mid x) := \underset{\pi}{\operatorname{argmin}} \, \mathbb{E}_{x,y\sim\pi(\cdot\mid x)}\left[\widehat{r}_\rho(x,y)\right] + \beta\mathbb{D}_{\text{KL}}(\pi \,\|\, \pi_\rho). \tag{8}$$

Then minimizing $\mathbb{E}_{x,y\sim\pi_{\text{ref}}}[\log \pi_\theta(y|x)]$ decreases the likelihood of responses sampled from $\pi_\rho^{\min}$:

$$\min_{\pi_\theta} \mathbb{E}_{x,y\sim\pi_{\text{ref}}(\cdot|x)}\left[\log \pi_\theta(y \mid x)\right] = \min_{\pi_\theta} \mathbb{E}_{x,y\sim\pi_\rho^{\min}(\cdot|x)}\left[\log \pi_\theta(y \mid x)\right].$$

The proofs for theorems in this section can be found in Appendix B and C. The above theorem states that maximizing the divergence between $\pi_\theta$ and $\pi_{\text{ref}}$ is essentially reducing the probability of generating responses with low implicit rewards reparameterized by any policy parameter $\rho$ during training. In other words, the LLM policy not only exploits the existing preference data but also learns to avoid generating the text $y$ that is assigned a low reward value. This process occurs in every iteration with updated reference models. Consequently, responses with high potential rewards are selectively preferred and many commonplace responses receive a small probability mass, thus mitigating the indiscriminate favoring of unseen responses and improving the exploration efficiency. In the next section, we will formally prove that the self-exploration mechanism is sample-efficient.

### 5.2 Self-Exploration is Provably Sample-Efficient

Following the proof technique of Xie et al. (2024), we provide the sample efficiency of the self-exploration mechanism by establishing a sublinear cumulative regret. Specifically, the cumulative regret $\mathcal{R}(T)$ up to $T$ iterations is defined as the cumulative performance discrepancy between the learned policy $\pi_t$ at iteration $t$ and the optimal policy $\pi^*$ (with respect to the regularized objective in (4)) over the run of the algorithm:

$$\mathcal{R}(T) = \sum_{t=1}^{T}[\mathcal{J}(\pi^*) - \mathcal{J}(\pi_t)].$$

The key idea is a reduction technique from Xie et al. (2024), which connects the sample complexity of SELM to that of existing RL algorithms (Zhong et al., 2022; Liu et al., 2024b). It is worth noting that the theoretical version of the self-exploration mechanism (Algorithm 2) is a bit different from the practical one used in the numerical experiments and is closer to the proposed algorithm in Xie et al. (2024).

**Assumption 5.2** (Realizable Policy Class with Regularity Condition)**.** We assume access to a finite policy class $\Pi$ containing the optimal $\pi^*$. Moreover, we assume that for any $\pi \in \Pi$ and prompt-response pair $(x,y)$,

$$\left|\log \frac{\pi(y \mid x)}{\pi_{\text{ref}}(y \mid x)}\right| \leq R_{\max}.$$

Assumption 5.2 stipulates that $\Pi$ is sufficiently comprehensive to include the optimal policy. Additionally, it imposes a bounded condition on $\log(\pi/\pi_{\text{ref}})$, which has been identified as the implicit reward function for DPO (Rafailov et al., 2024b). It naturally extends to the infinite case by introducing concepts like the covering number. An example of an infinite policy class is the log-linear policy class, whose complexity is approximately $d$ (see, e.g., Zanette et al. (2021) for a detailed derivation).

**Theorem 5.3.** Under Assumption 5.2, let $\eta = \sqrt{Td_{\text{PGEC}}/(\exp(4R_{\max})\log(|\Pi|/\delta))}$, $\alpha = 2/(\eta\exp(4R_{\max}))$, and $\delta \in (0,1)$. Then with probability at least $1-\delta$, we have

$$\mathcal{R}(T) \lesssim \sqrt{d_{\text{PGEC}} \cdot \exp(2R_{\max}) \cdot T \cdot \log(|\Pi|/\delta)},$$

where $\lesssim$ omits absolute constants, and $d_{\mathrm{PGEC}}$ is a preference-based version of Generalized Eluder Coefficient (GEC; Zhong et al., 2022) defined in Appendix C.1 capturing the complexity of learning problem. For log-linear policy class $\Pi = \{\pi_\theta : \pi_\theta(y \,|\, x) \propto \exp(\langle\phi(x,y),\theta\rangle/\beta)\}$ with $d$-dimensional feature $\phi$, it holds that $d_{\mathrm{PGEC}} \le \widetilde{O}(d)$.

The proof technique is from Xie et al. (2024), which connects RLHF with RL and allows us to use the preference-based version of GEC (Zhong et al., 2022; Liu et al., 2024b) as the complexity measure to characterize the cumulative regret $\mathcal{R}(T)$. We restate the proof technique from Xie et al. (2024) for completeness. We emphasize that it is not a novel contribution of the present work. Since the cumulative regret is sublinear in the number of iterations $T$, the above theorem indicates that the policy $\pi_t$ converges to the optimal $\pi^*$ within sufficient iterations. Moreover, by the standard online-to-batch argument, Theorem 5.3 shows that SELM is capable of finding an $\varepsilon$-optimal policy with a sample complexity of $\widetilde{O}(1/\varepsilon^2)$. This highlights the sample efficiency of SELM from the theoretical perspective.

# 6 Experiments

## 6.1 Experiment Setup

We adopt UltraFeedback (Cui et al., 2023) as our training dataset, which contains 61k preference pairs of single-turn conversations. For the external ranker during online alignment, we choose the small-sized PairRM (0.4B) (Jiang et al., 2023). All experiments are conducted on 8xA100 GPUs.

Due to the absence of performant open-source online direct alignment codebases at the time of this study, we first implement an iterative version of DPO as the baseline, adhering to the same steps as Algorithm 1 but training the LLM with the standard DPO objective. Then we conduct a grid search over hyperparameters, such as the batch size, learning rate, and iteration number, to identify the optimal settings for the iterative DPO baseline. We follow these best settings to train SELM. In addition, we apply iterative DPO and SELM on instruction fine-tuned models. Specifically, we consider two series of LLMs: Zephyr (Tunstall et al., 2023b) and Llama-3 (Meta, 2024), to demonstrate the robustness of SELM. Since the official Zephyr-7B-$\beta$ model is fine-tuned with DPO on the same UltraFeedback dataset, to avoid overoptimization, we choose Zephyr-7B-SFT[1] as the base model and perform 3 iterations of SELM after a single iteration of standard DPO training on the first portion of the training data (we refer to this model as Zephyr-7B-DPO). For Llama-3-8B-Instruct[2] that is already fine-tuned with RLHF, we directly apply 3 iterations of SELM training.

## 6.2 Experiment Results

We first report the performance of SELM and the baselines on the instruction-following chat benchmarks AlpacaEval 2.0 (Dubois et al., 2024) and MT-Bench (Zheng et al., 2024) in Table 1. We can observe that for AlpacaEval 2.0, SELM significantly boosts Zephyr-7B-SFT and Llama-3-8B-Instruct, achieving length-controlled (LC) win rate improvements of $+16.24\%$ and $+11.75\%$, respectively. This enhancement results in models that are competitive with or even superior to much larger LLMs, such as Yi-34B-Chat (Young et al., 2024) and Llama-3-70B-Instruct. For the multi-turn MT-Bench, which exhibits higher variance, we report the average scores of SELM and DPO baselines across 3 runs. We observe that SELM improves the scores by $+2.31$ and $+0.32$, respectively. Furthermore, the proposed method self-explores and enhances the model monotonically, with consistent performance improvements in each iteration. This validates the robustness of our algorithm. Compared to other iterative post-training algorithms, such as SPIN (Chen et al., 2024), DNO (Rosset et al., 2024), and SPPO (Wu et al., 2024), SELM gains more improvements on both

---

[1]https://huggingface.co/HuggingFaceH4/mistral-7b-sft-beta
[2]https://huggingface.co/meta-llama/Meta-Llama-3-8B-Instruct

benchmarks when using the weaker base model (Zephyr-7B-SFT), and achieves the best performance when using Llama-3-8B-Instruct as the base model.

| Model | AlpacaEval 2.0 | | | MT-Bench | | |
|---|---|---|---|---|---|---|
| | LC Win Rate | Win Rate | Avg. len | Avgerage | 1st Turn | 2nd Turn |
| Zephyr-7B-SFT | 8.01 | 4.63 | 916 | 5.30 | 5.63 | 4.97 |
| Zephyr-7B-DPO | 15.41 | 14.44 | 1752 | 7.31 | 7.55 | 7.07 |
| DPO Iter 1 (Zephyr) | 20.53 | 16.69 | 1598 | 7.53 | 7.81 | 7.25 |
| DPO Iter 2 (Zephyr) | 22.12 | 19.82 | 1717 | 7.55 | 7.85 | 7.24 |
| DPO Iter 3 (Zephyr) | 22.19 (↑14.18) | 19.88 | 1717 | 7.46 (↑2.16) | 7.85 | 7.06 |
| SELM Iter 1 (Zephyr) | 20.52 | 17.23 | 1624 | 7.53 | 7.74 | 7.31 |
| SELM Iter 2 (Zephyr) | 21.84 | 18.78 | 1665 | 7.61 | **7.85** | 7.38 |
| SELM Iter 3 (Zephyr) | **24.25**(↑16.24) | **21.05** | 1694 | **7.61** (↑2.31) | 7.74 | **7.49** |
| Llama-3-8B-Instruct | 22.92 | 22.57 | 1899 | 7.93 | 8.47 | 7.38 |
| DPO Iter 1 (Llama3-It) | 30.89 | 31.60 | 1979 | 8.07 | 8.44 | 7.70 |
| DPO Iter 2 (Llama3-It) | 33.91 | 32.95 | 1939 | 7.99 | 8.39 | 7.60 |
| DPO Iter 3 (Llama3-It) | 33.17 (↑10.25) | 32.18 | 1930 | 8.18 (↑0.25) | 8.60 | 7.77 |
| SELM Iter 1 (Llama3-It) | 31.09 | 30.90 | 1956 | 8.09 | 8.57 | 7.61 |
| SELM Iter 2 (Llama3-It) | 33.53 | 32.61 | 1919 | 8.18 | **8.69** | 7.66 |
| SELM Iter 3 (Llama3-It) | **34.67** (↑11.75) | **34.78** | 1948 | **8.25** (↑0.32) | 8.53 | **7.98** |
| SPIN | 7.23 | 6.54 | 1426 | 6.54 | 6.94 | 6.14 |
| Orca-2.5-SFT | 10.76 | 6.99 | 1174 | 6.88 | 7.72 | 6.02 |
| DNO (Orca-2.5-SFT) | 22.59 | 24.97 | 2228 | 7.48 | 7.62 | 7.35 |
| Mistral-7B-Instruct-v0.2 | 19.39 | 15.75 | 1565 | 7.51 | 7.78 | 7.25 |
| SPPO (Mistral-it) | 28.53 | 31.02 | 2163 | 7.59 | 7.84 | 7.34 |
| Yi-34B-Chat | 27.19 | 21.23 | 2123 | 7.90 | - | - |
| Llama-3-70B-Instruct | 33.17 | 33.18 | 1919 | 9.01 | 9.21 | 8.80 |
| GPT-4 Turbo (04/09) | 55.02 | 46.12 | 1802 | 9.19 | 9.38 | 9.00 |

Table 1: Results on AlpacaEval 2.0 and MT-Bench averaged with 3 runs. Names inside the brackets are the models that are aligned based upon. The red arrows indicate the increment or decrement from the base model. Compared to iterative DPO and other online alignment baselines, SELM gains more improvements based on the weaker Zephyr-7B-SFT model and achieves superior performance that is competitive with much larger SOTA models when fine-tuned from Llama-3-8B-Instruct.

Notably, the implemented iterative DPO is obtained through comprehensive grid searches of hyperparameters and practical designs (see Appendix D for details), making it a strong baseline comparable with SOTA online alignment algorithms fine-tuned from more advanced models, such as SPIN, DNO, and SPPO. For example, DPO Iter 3 (Zephyr) achieves an MT-Bench score of 7.46, representing a 2.16 improvement over Zephyr-SFT (5.30) and coming close to DNO (7.48), which is fine-tuned from the stronger model Orca-2.5-SFT (6.88). Additionally, SPPO achieves an MT-Bench score of 7.59, a modest improvement of 0.08 over Mistral-it (7.51).

We also conduct pairwise comparisons between SELM, iterative DPO, and the base models to validate the effectiveness of our method. The results for AlpacaEval 2.0 are shown in Figure 2. We observe that with the same number of training iterations and data, SELM consistently outperforms the iterative DPO counterpart. Additionally, when using Zephyr-7B-SFT as the base model, SELM outperforms iterative DPO even when the latter is trained with twice the data.

Beyond instruction-following benchmarks, we also evaluate SELM and the baselines on several academic benchmarks, including GSM8K (Cobbe et al., 2021), HellaSwag (Zellers et al., 2019), ARC challenge (Clark et al., 2018), TruthfulQA (Lin et al., 2021), EQ-Bench (Paech, 2023), and OpenBookQA (OBQA) (Mihaylov et al., 2018). To better reflect the capabilities of LLMs, we adopt various settings for these benchmarks, including zero-shot, few-shot, and few-shot Chain-of-Thought (CoT) settings. The accuracy results for these

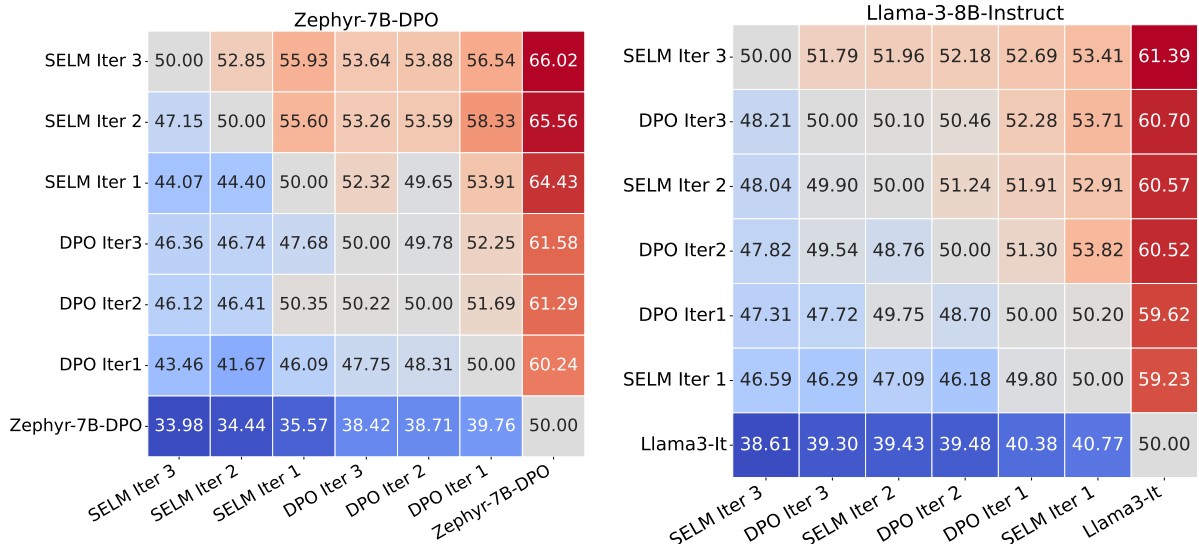

Figure 2: Pairwise comparison between SELM, iterative DPO, and base models. Scores represent the LC win rates of the row models against the column models. Models positioned in higher rows have higher LC win rates against the base model and thus better performance.

multiple-choice QA benchmarks are provided in Table 2. It can be observed that both our method and the baselines can degrade after the RLHF phase on some benchmarks, which is known as the alignment tax (Askell et al., 2021; Noukhovitch et al., 2024; Li et al., 2024b). Nevertheless, our method is still able to improve the base models on most of the benchmarks and offers the best overall performance.

We note that SELM is one of the instantiations of the proposed self-exploration objective in (1), with reparameterized reward functions and algorithm-specific designs described in Section 4.2, such as the dataset partition and update rule. However, this objective is not restricted to the current implementation and can also be directly applied to any other online alignment framework, with or without a separate reward model, regardless of differences in algorithm designs. Thus, the proposed method is orthogonal to and can be integrated directly into the recent online RLHF workflows (Dong et al., 2024; Xiong et al., 2023; Hu et al., 2024) that incorporate additional delicate designs with carefully curated datasets.

## 6.3 Ablation Study

We first provide ablation studies to better understand the explorative optimism term. We begin by investigating the effect of the optimism coefficient $\alpha$. In Figure 3 (Left), we plot the LC win rates of SELM when using Zephyr-7B-SFT as the base model for different $\alpha$ in the AlpacaEval 2.0 benchmark. We find that setting a small $\alpha$, such as 0.0001, leads to very similar behaviors to the iterative DPO ($\alpha = 0$) baseline, while SELM with a large $\alpha$ may become overly optimistic and thus not very effective. These results meet our expectations, suggesting that proper values of $\alpha$ are essential for achieving the best trade-off between exploration and exploitation.

Next, we study the difference in reward distributions with varied $\alpha$ and iterations. Specifically, for prompts from the 2k test set of UltraFeedback, we greedily sample from the LLM and generate rewards for the responses with PairRM. We then calculate the fraction of data that lies in each partition of rewards. The results for different $\alpha$ values of SELM Iter 2 (Zephyr) in Figure 3 (Middle) indicates that increasing $\alpha$ results in distributions that are concentrated in higher-reward regions.

| Models | GSM8K (8-s CoT) | HellaSwag (10-s) | ARC (25-s) | TruthfulQA (0-s) | EQ (0-s) | OBQA (10-s) | Average |
|---|---|---|---|---|---|---|---|
| Zephyr-7B-SFT | 43.8 | 82.2 | 57.4 | 43.6 | 39.1 | 35.4 | 50.3 |
| Zephyr-7B-DPO | 47.2 | 84.5 | 61.9 | 45.5 | 65.2 | 38.0 | 57.0 |
| DPO Iter 1 (Zephyr) | 45.5 | 85.2 | 62.1 | 52.4 | 68.4 | 39.0 | 58.8 |
| DPO Iter 2 (Zephyr) | 44.9 | 85.4 | 62.0 | 53.1 | 69.3 | 39.4 | 59.0 |
| DPO Iter 3 (Zephyr) | 43.2 | 85.2 | 60.8 | 52.5 | 69.1 | 39.6 | 58.4 |
| SELM Iter 1 (Zephyr) | 46.3 | 84.8 | 62.9 | 52.9 | 68.8 | 39.6 | 59.2 |
| SELM Iter 2 (Zephyr) | 46.2 | 85.4 | 62.1 | 53.1 | 69.3 | 39.6 | 59.3 |
| SELM Iter 3 (Zephyr) | 43.8 | 85.4 | 61.9 | 52.4 | 69.9 | 39.8 | 58.9 |
| Llama-3-8B-Instruct | 76.7 | 78.6 | 60.8 | 51.7 | 61.8 | 38.0 | 61.3 |
| DPO Iter 1 (Llama3-It) | 78.5 | 81.7 | 63.9 | 55.5 | 64.1 | 42.6 | 64.4 |
| DPO Iter 2 (Llama3-It) | 79.4 | 81.7 | 64.4 | 56.4 | 64.3 | 42.6 | 64.8 |
| DPO Iter 3 (Llama3-It) | 80.1 | 81.7 | 64.1 | 56.5 | 64.1 | 42.6 | 64.8 |
| SELM Iter 1 (Llama3-It) | 78.7 | 81.7 | 64.5 | 55.4 | 64.1 | 42.4 | 64.5 |
| SELM Iter 2 (Llama3-It) | 79.3 | 81.8 | 64.7 | 56.5 | 64.2 | 42.6 | 64.9 |
| SELM Iter 3 (Llama3-It) | 80.1 | 81.8 | 64.3 | 56.5 | 64.2 | 42.8 | 65.0 |
| SPIN | 44.7 | 85.9 | 65.9 | 55.6 | 54.4 | 39.6 | 57.7 |
| Mistral-7B-Instruct-v0.2 | 43.4 | 85.3 | 63.4 | 67.5 | 65.9 | 41.2 | 61.1 |
| SPPO (Mistral-it) | 42.4 | 85.6 | 65.4 | 70.7 | 56.5 | 40.0 | 60.1 |

Table 2: Performance comparison between SELM and the baselines on academic multi-choice QA benchmarks in standard zero-shot, few-shot, and CoT settings. Here, n-s refers to n-shot. The red and blue texts represent the best and the second-best results.

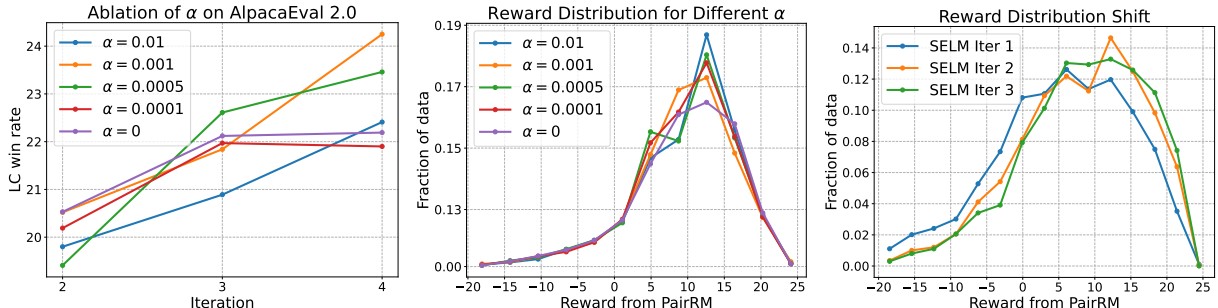

Figure 3: Ablation on the optimism coefficient $\alpha$ and the change of the reward distribution. **Left:** The length-controlled win rates of SELM with different $\alpha$ on AlpacaEval 2.0. **Middle:** Comparison of reward distributions at iteration 2 with different $\alpha$. **Right:** SELM initially explores and then shifts to higher-reward regions as more training iterations are performed.

Additionally, Figure 3 (Right) demonstrates that the reward distribution shifts to the right (higher) as more training iterations are performed. This shift corresponds to an initial exploration phase, where the LLM generates uncertain responses of varying quality, followed by an exploitation phase as feedback is incorporated and more training data is collected.

We also conduct ablation studies on the implicit reward captured by the SELM and DPO models. Recall that for both SELM and DPO, the implicit reward takes the form of $\widehat{r}_\theta(x, y) = \beta(\log \pi_\theta(y \mid x) - \log \pi_{\text{ref}}(y \mid x))$. We calculate the reward difference $\widehat{r}_{\text{SELM}}(x, y) - \widehat{r}_{\text{DPO}}(x, y)$ for each prompt $x$ in the UltraFeedback holdout test set. Here, we study the implicit reward of the good (chosen) and bad (rejected) responses, so $y = y_w$ or $y = y_l$. We then sort the reward difference and plot the results for Zephyr-based models after iteration 1 in Figure 4. The plot clearly shows that for both chosen and rejected responses, SELM produces higher *implicit* rewards compared to DPO, aligning with the proposed optimistically biased self-exploration objective.

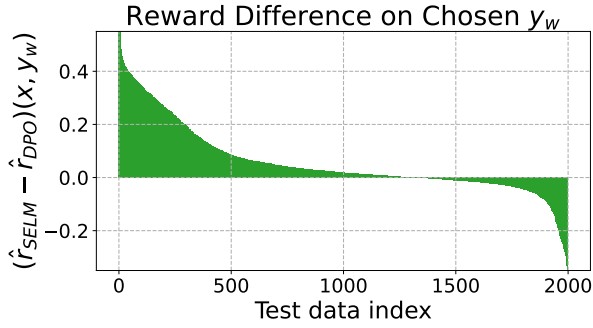 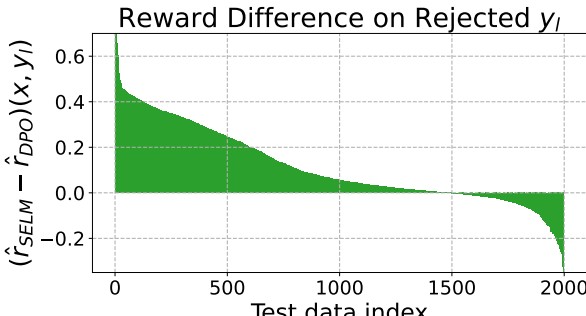

Figure 4: Difference of implicit reward between SELM and DPO on the chosen and rejected responses. SELM assigns a higher implicit reward than DPO for both responses.

In Section 5, we show that SELM engages in more active exploration by prioritizing high-reward responses compared to DPO, which indiscriminately favors unseen extrapolations and explores passively. To validate this, we sample three responses from SELM and DPO Iter 2 (Zephyr) for each prompt and we calculate the subtraction of the mean implicit rewards. As illustrated in Figure 5, SELM consistently achieves higher implicit rewards across most prompts, with the positive reward differences being notably larger in magnitude, supporting our claim regarding SELM's active exploration behavior. We refer the readers to Appendix E for more ablation studies, including the effect of prompts across iterations, preference pair construction, and optimism bias versus data selection.

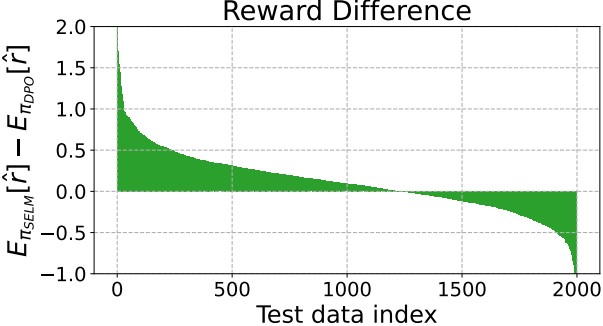

Figure 5: SELM actively explores by favoring high-reward responses.

## 7 Conclusion & Future Work

In this paper, we introduced an active preference elicitation method for the online alignment of large language models. By incorporating an optimism term into the reward-fitting objective, the proposed bilevel self-exploring objective effectively balances between exploiting observed data and exploring potentially high-reward regions. Unlike standard online RLHF algorithms that passively explore the response space by sampling from the training LLM, whose sole objective is maximizing the expected learned reward, our method actively seeks diverse and high-quality responses. This self-exploration mechanism helps mitigate the risk of premature convergence and overfitting when the reward model is only locally accurate. To optimize this bilevel objective, we solve the inner-level problem and reparameterize the reward with the LLM policy, resulting in a simple yet novel iterative alignment algorithm called *Self-Exploring Language Models* (SELM). Compared to DPO, SELM is provably sample-efficient and improves the exploration efficiency by selectively favoring responses with high potential rewards rather than indiscriminately sampling unseen responses.

Our experiments, conducted with Zephyr-7B-SFT and Llama-3-8B-Instruct models, demonstrate the efficacy of SELM with consistent improvements on AlpacaEval 2.0, MT-Bench, and academic benchmarks. These results underscore the ability of SELM to enhance the alignment and capabilities of LLMs by promoting more diverse and high-quality responses. Since the proposed technique is orthogonal to the adopted online RLHF workflow, it will be interesting to apply our method within more sophisticated alignment frameworks with advanced designs, which we would like to leave as future work.

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

## A    Derivations in Section 4.1

We begin by deriving (6). The solution for the inner-level optimization problem of (5) is as follows:

$$
\max_{\pi} \mathcal{F}(\pi; r) = \max_{\pi} \mathbb{E}_{\substack{x\sim\mathcal{D}_t, y\sim\pi(\cdot|x) \\ y'\sim\pi_{\text{ref}}(\cdot|x)}} \Big[ r(x,y) - r(x,y') \Big] - \beta \mathbb{D}_{\text{KL}}(\pi \,||\, \pi_{\text{ref}})
$$
$$
= \mathbb{E}_{x\sim\mathcal{D}_t} \Big[ \beta \log \mathbb{E}_{y\sim\pi_{\text{ref}}(\cdot|x)} \big[ \exp(r(x,y)/\beta) \big] \Big] - \mathbb{E}_{x\sim\mathcal{D}_t, y'\sim\pi_{\text{ref}}(\cdot|x)} \big[ r(x,y') \big] \tag{9}
$$

When the reward $r$ is reparameterized by $\widehat{r}_\theta(x,y) = \beta(\log \pi_\theta(y \mid x) - \log \pi_{\text{ref}}(y \mid x))$, we have that the first term in (9) is 0. The bilevel objective (5) then becomes

$$
\max_{r} -\mathcal{L}_{\text{lr}}(r; \mathcal{D}_t) - \alpha \mathbb{E}_{x\sim\mathcal{D}, y'\sim\pi_{\text{ref}}(\cdot|x)} \big[ r(x,y') \big].
$$

By reparameterizing the reward with the LLM, we obtain the desired results in (6).

Then we provide the derivation of (7). We primarily consider the gradient of the newly incorporated term $\mathbb{E}_{x\sim\mathcal{D}, y\sim\pi_{\text{ref}}(\cdot|x)}[\log \pi_\theta(y \mid x)]$. Specifically, we have

$$
\nabla_\theta \mathbb{E}_{x\sim\mathcal{D}, y\sim\pi_{\text{ref}}(\cdot|x)} \big[ \log \pi_\theta(y \mid x) \big] = \mathbb{E}_{x\sim\mathcal{D}} \Big[ \sum_{y} \pi_{\text{ref}}(y \mid x) \nabla_\theta \log \pi_\theta(y \mid x) \Big]
$$
$$
= \mathbb{E}_{x\sim\mathcal{D}, y\sim\pi_\theta} \Big[ \frac{\pi_{\text{ref}}(y \mid x)}{\pi_\theta(y \mid x)} \nabla_\theta \log \pi_\theta(y \mid x) \Big]
$$
$$
= \mathbb{E}_{x\sim\mathcal{D}, y\sim\pi_\theta} \Big[ \exp\big(-\widehat{r}_\theta(x.y)/\beta\big) \nabla_\theta \log \pi_\theta(y \mid x) \Big].
$$

For the derivation of the DPO gradient $\nabla_\theta \mathcal{L}_{\text{DPO}}(\pi_\theta; \mathcal{D}_t)$, we refer the readers to Rafailov et al. (2024b).

## B    Proof of Theorem 5.1

*Proof of Theorem 5.1.* The solution to the KL-constrained reward minimization objective (8) is

$$
\pi_\rho^{\min}(y \mid x) = \pi_\rho(y \mid x) \exp\big(-\widehat{r}_\rho(x,y)/\beta\big)/Z_r(x),
$$

where $Z_r(x) = \sum_y \pi_\rho(y \mid x) \exp(-\widehat{r}_\rho(x,y)/\beta) = 1$. Then we have $\pi_\rho^{\min}(y \mid x) = \pi_{\text{ref}}(y \mid x)$, i.e., the reference policy $\pi_{\text{ref}}$ achieves the lowest implicit reward reparameterized by any $\rho$. $\square$

# C   Proof of Theorem 5.3

We use the reduction technique from Xie et al. (2024) to connect the sample complexity of SELM to that of existing RL algorithms (Zhong et al., 2022; Liu et al., 2024b). We restate the proof technique from Xie et al. (2024) for completeness. We emphasize that it is not a novel contribution of the present work. It is worth noting that the theoretical version of the self-exploration mechanism (Algorithm 2) is a bit different from the practical one used in the numerical experiments and is closer to the proposed algorithm in Xie et al. (2024).

We present the following theoretical version of the proposed self-exploration algorithm. The key modification in Algorithm 1 lies in its pragmatic strategy for constructing the chosen and rejected responses. Despite this adjustment, the core principles of leveraging the self-exploration objective during online alignment remain the same.

---

**Algorithm 2** Self-Exploring Language Models (SELM; Theoretical Version)

---

**Input:** Reference model $\pi_{\text{ref}}$, preference dataset $\mathcal{D}_0 = \varnothing$, prompt distribution $\nu$, online iterations $T$, optimism coefficient $\alpha$, $\pi_0 = \pi_{\text{ref}}$.

1: **for** iteration $t = 1, 2, \ldots, T$ **do**
2:      Sample $x_t \sim \nu$, $y_t^1 \sim \pi_{t-1}(\cdot \mid x)$, $y_t^2 \sim \pi_{t-1}(\cdot \mid x)$.
3:      Update the preference data $\mathcal{D}_t = \mathcal{D}_{t-1} \cup \{(x_t, y_t^1, y_t^2)\}$
4:      Train the LLM $\pi_t = \operatorname{argmax}_\pi\{-\mathcal{L}_{\text{DPO}}(\pi; \mathcal{D}_t) - \alpha \cdot \mathbb{E}_{x \sim \nu}\mathbb{E}_{y \sim \pi_{\text{ref}}(\cdot \mid x)}[\log \pi(y \mid x)]\}$, let $\pi_{\text{ref}} = \pi_t$.
5: **end for**

---

**Definition C.1** (Preference-based GEC). For the function class $\Pi$, we define the preference-based GEC (PGEC) as the smallest $d_{\text{GPEC}}$ as

$$
\sum_{t=1}^{T} \mathbb{E}_{(x,y,y') \sim (\nu, \pi_{\text{ref}}, \pi_t)} \left[ \log \frac{\pi^*(y \mid x)}{\pi_{\text{ref}}(y \mid x)} - \log \frac{\pi_t(y \mid x)}{\pi_{\text{ref}}(y \mid x)} - \log \frac{\pi^*(y' \mid x)}{\pi_{\text{ref}}(y' \mid x)} + \log \frac{\pi_t(y' \mid x)}{\pi_{\text{ref}}(y' \mid x)} \right]
$$

$$
\leq \sqrt{d_{\text{PGEC}} \sum_{t=1}^{T} \sum_{\tau=1}^{t-1} \mathbb{E}_{(x,y,y') \sim (\nu, \pi_{\text{ref}}, \pi^\tau)} \left[ \log \frac{\pi^*(y \mid x)}{\pi_{\text{ref}}(y \mid x)} - \log \frac{\pi^\tau(y \mid x)}{\pi_{\text{ref}}(y \mid x)} - \log \frac{\pi^*(y' \mid x)}{\pi_{\text{ref}}(y' \mid x)} + \log \frac{\pi^\tau(y' \mid x)}{\pi_{\text{ref}}(y' \mid x)} \right]^2}
$$

$$
+ 4\sqrt{d_{\text{PGEC}} T}.
$$

The definition of PGEC is a preference-based version of Generalized Eluder Coefficient (GEC) proposed by (Zhong et al., 2022). Intuitively, both PGEC and GEC establish a crucial connection between *prediction error* and *in-sample estimation error*, effectively transforming regret minimization into an online estimation problem. For a comprehensive explanation and in-depth discussion, readers are directed to Zhong et al. (2022). A slight difference is that the PGEC here is defined with respect to the policy class, while the GEC in Zhong et al. (2022) is defined with respect to the model or value class. These can be connected if we regard the implicit reward class $\log(\pi/\pi_{\text{ref}})$ as the model or value class. As an important example, if we consider the log-linear function class $\Pi = \{\pi_\theta : \pi_\theta(y \mid x) \propto \exp(\langle \phi(x,y), \theta \rangle / \beta)\}$, we can show that $d_{\text{PGEC}} = \widetilde{O}(d)$ by the elliptical potential lemma (Abbasi-Yadkori et al., 2011; Zhong et al., 2022). Another remark is that here the PGEC is defined in the bandit formulation, and it can be naturally extended to the token-wise MDP formulation (Zhong et al., 2024; Rafailov et al., 2024a; Xie et al., 2024) and further connects to the eluder dimension in the context of preference-based MDPs (Chen et al., 2022; Wang et al., 2023). Specifically, if we regard the generation process of LLMs as token-level MDPs where the generation of each token serves as one step, the learning objective is maximizing

$$
\mathcal{J}(\pi) = \mathbb{E}_{x \sim \nu, \tau \sim \pi} \left[ r(\tau) - \beta \log \frac{\pi(\tau \mid x)}{\pi_{\text{ref}}(\tau \mid x)} \right].
$$

Here $\tau$ is the full trajectory starting from $x$. We can similarly define the PGEC (Definition C.1) for token-wise MDPs by replacing the response $y, y'$ in the bandit formulation with the trajectories $\tau, \tau'$ in the token-wise MDP formulation. We have the following informal theorem:

**Theorem C.2** (Regret for MDP Formulation (informal))**.** With proper parameter choice, it holds with probability at least $1 - \delta$ that

$$\mathcal{R}(T) \lesssim \sqrt{d_{\text{PGEC}} \cdot \exp(2V_{\max}) \cdot T \cdot \log(|\Pi|/\delta)},$$

where $V_{\max}$ is a bounded coefficient for token-wise MDPs, similar to the one described in Assumption 5.2.

## C.1 Proof of Theorem 5.3

*Proof of Theorem 5.3.* We first decompose the regret as

$$
\begin{aligned}
\mathcal{R}(T) &= \sum_{t=1}^{T} [\mathcal{J}(\pi^*) - \mathcal{J}(\pi_t)] \\
&= \sum_{t=1}^{T} \left( \mathbb{E}_{x \sim \nu, y \sim \pi^*(\cdot|x)} \left[ r(x,y) - \beta \log \frac{\pi^*(y \mid x)}{\pi_{\text{ref}}(y \mid x)} \right] - \mathbb{E}_{x \sim \nu, y \sim \pi_t(\cdot|x)} \left[ r(x,y) - \beta \log \frac{\pi_t(y \mid x)}{\pi_{\text{ref}}(y \mid x)} \right] \right) \\
&= \sum_{t=1}^{T} \left( \mathbb{E}_{x \sim \nu, y \sim \pi_{\text{ref}}(\cdot|x)} \left[ r(x,y) - \beta \log \frac{\pi^*(y \mid x)}{\pi_{\text{ref}}(y \mid x)} \right] - \mathbb{E}_{x \sim \nu, y \sim \pi_t(\cdot|x)} \left[ r(x,y) - \beta \log \frac{\pi_t(y \mid x)}{\pi_{\text{ref}}(y \mid x)} \right] \right),
\end{aligned}
$$

where the last line uses the fact that

$$r(x,y) - \beta \log \frac{\pi^*(y \mid x)}{\pi_{\text{ref}}(y \mid x)} = \beta \cdot \log Z_r(x), \tag{10}$$

which is independent of the response $y$. As the optimal solution to the KL-constrained reward maximization objective, the proof of (10) can be found in many previous works (Rafailov et al., 2024b; Peng et al., 2019), e.g., see Appendix A.1 in Rafailov et al. (2024b). Rearranging the above regret decomposition, we have

$$
\begin{aligned}
\mathcal{R}(T) &= \sum_{t=1}^{T} \left( \mathbb{E}_{x \sim \nu, y \sim \pi_{\text{ref}}(\cdot|x)} \left[ r(x,y) - \beta \log \frac{\pi^*(y \mid x)}{\pi_{\text{ref}}(y \mid x)} \right] - \mathbb{E}_{x \sim \nu, y \sim \pi_t(\cdot|x)} \left[ r(x,y) - \beta \log \frac{\pi_t(y \mid x)}{\pi_{\text{ref}}(y \mid x)} \right] \right) \\
&= \sum_{t=1}^{T} \mathbb{E}_{x \sim \nu, y \sim \pi_{\text{ref}}(\cdot|x)} \left[ \beta \log \frac{\pi_t(y \mid x)}{\pi^*(y \mid x)} \right] \\
&\quad + \sum_{t=1}^{T} \mathbb{E}_{x \sim \nu, y \sim \pi_{\text{ref}}(\cdot|x), y' \sim \pi_t(\cdot|x)} \left[ r(x,y) - \beta \log \frac{\pi_t(y \mid x)}{\pi_{\text{ref}}(y \mid x)} - r(x,y') + \beta \log \frac{\pi_t(y' \mid x)}{\pi_{\text{ref}}(y' \mid x)} \right] \\
&= \sum_{t=1}^{T} \mathbb{E}_{x \sim \nu, y \sim \pi_{\text{ref}}(\cdot|x)} \left[ \beta \log \frac{\pi_t(y \mid x)}{\pi^*(y \mid x)} \right] \\
&\quad + \beta \sum_{t=1}^{T} \mathbb{E}_{(x,y,y') \sim (\nu, \pi_{\text{ref}}, \pi_t)} \left[ \log \frac{\pi^*(y \mid x)}{\pi_{\text{ref}}(y \mid x)} - \log \frac{\pi_t(y \mid x)}{\pi_{\text{ref}}(y \mid x)} - \log \frac{\pi^*(y' \mid x)}{\pi_{\text{ref}}(y' \mid x)} + \log \frac{\pi_t(y' \mid x)}{\pi_{\text{ref}}(y' \mid x)} \right], \tag{11}
\end{aligned}
$$

where the last line uses (10). By the definition of PGEC in Definition C.1, we have

$$
\sum_{t=1}^{T} \mathbb{E}_{(x,y,y')\sim(\nu,\pi_{\mathrm{ref}},\pi_t)} \left[ \log \frac{\pi^*(y\,|\,x)}{\pi_{\mathrm{ref}}(y\,|\,x)} - \log \frac{\pi_t(y\,|\,x)}{\pi_{\mathrm{ref}}(y\,|\,x)} - \log \frac{\pi^*(y'\,|\,x)}{\pi_{\mathrm{ref}}(y'\,|\,x)} + \log \frac{\pi_t(y'\,|\,x)}{\pi_{\mathrm{ref}}(y'\,|\,x)} \right]
$$

$$
\leq \sqrt{ d_{\mathrm{PGEC}} \sum_{t=1}^{T}\sum_{\tau=1}^{t-1} \mathbb{E}_{(x,y,y')\sim(\nu,\pi_{\mathrm{ref}},\pi^\tau)} \left[ \log \frac{\pi^*(y\,|\,x)}{\pi_{\mathrm{ref}}(y\,|\,x)} - \log \frac{\pi^\tau(y\,|\,x)}{\pi_{\mathrm{ref}}(y\,|\,x)} - \log \frac{\pi^*(y'\,|\,x)}{\pi_{\mathrm{ref}}(y'\,|\,x)} + \log \frac{\pi^\tau(y'\,|\,x)}{\pi_{\mathrm{ref}}(y'\,|\,x)} \right]^2 }
$$

$$
+ 4\sqrt{d_{\mathrm{PGEC}}T}
$$

$$
\leq \frac{d_{\mathrm{PGEC}}}{4\eta} + \eta \sum_{t=1}^{T}\sum_{\tau=1}^{t-1} \mathbb{E}_{(x,y,y')\sim(\nu,\pi_{\mathrm{ref}},\pi^\tau)} \left[ \log \frac{\pi^*(y\,|\,x)}{\pi_{\mathrm{ref}}(y\,|\,x)} - \log \frac{\pi^\tau(y\,|\,x)}{\pi_{\mathrm{ref}}(y\,|\,x)} - \log \frac{\pi^*(y'\,|\,x)}{\pi_{\mathrm{ref}}(y'\,|\,x)} + \log \frac{\pi^\tau(y'\,|\,x)}{\pi_{\mathrm{ref}}(y'\,|\,x)} \right]^2,
$$

$$
+ 4\sqrt{d_{\mathrm{PGEC}}T}, \tag{12}
$$

where the last inequality follows from the fact that $\sqrt{xy} \leq x/(4\eta) + \eta y$ for any $x, y, \eta > 0$.

By the updating rule of $\pi_{t+1} = \mathrm{argmax}_\pi\{-\mathcal{L}_{\mathrm{DPO}}(\pi;\mathcal{D}_t) - \alpha \cdot \mathbb{E}_{x\sim\nu}\mathbb{E}_{y\sim\pi_{\mathrm{ref}}(\cdot|x)}[\log\pi(y\,|\,x)]\}$, we have

$$
-\mathcal{L}_{\mathrm{DPO}}(\pi_t;\mathcal{D}_{t-1}) - \alpha \cdot \mathbb{E}_{x\sim\nu,y\sim\pi_{\mathrm{ref}}(\cdot|x)}[\log\pi_t(y\,|\,x)]
$$

$$
\geq -\mathcal{L}_{\mathrm{DPO}}(\pi^*;\mathcal{D}_{t-1}) - \alpha \cdot \mathbb{E}_{x\sim\nu,y\sim\pi_{\mathrm{ref}}(\cdot|x)}[\log\pi^*(y\,|\,x)],
$$

which equivalents to that

$$
\mathbb{E}_{x\sim\nu,y\sim\pi_{\mathrm{ref}}(\cdot|x)} \left[ \beta \log \frac{\pi_t(y\,|\,x)}{\pi^*(y\,|\,x)} \right] \leq \frac{\beta}{\alpha} \cdot \left( \mathcal{L}_{\mathrm{DPO}}(\pi^*;\mathcal{D}_{t-1}) - \mathcal{L}_{\mathrm{DPO}}(\pi_t;\mathcal{D}_{t-1}) \right). \tag{13}
$$

We upper bound the right-hand side of (13) via the following lemma.

**Lemma C.3** (Concentration)**.** For any $t \in [T]$ and $0 < \delta < 1$, it holds with probability $1 - \delta$ that

$$
\mathcal{L}_{\mathrm{DPO}}(\pi^*;\mathcal{D}_{t-1}) - \mathcal{L}_{\mathrm{DPO}}(\pi_t;\mathcal{D}_{t-1})
$$

$$
\lesssim -\frac{2}{\exp(4R_{\max})} \cdot \sum_{\tau=1}^{t-1} \mathbb{E}_{(x,y,y')\sim(\nu,\pi_{\mathrm{ref}},\pi^\tau)} \left[ \log \frac{\pi^*(y\,|\,x)}{\pi_{\mathrm{ref}}(y\,|\,x)} - \log \frac{\pi^\tau(y\,|\,x)}{\pi_{\mathrm{ref}}(y\,|\,x)} - \log \frac{\pi^*(y'\,|\,x)}{\pi_{\mathrm{ref}}(y'\,|\,x)} + \log \frac{\pi^\tau(y'\,|\,x)}{\pi_{\mathrm{ref}}(y'\,|\,x)} \right]^2
$$

$$
+ \log(|\Pi|/\delta).
$$

*Proof.* The proof of this lemma follows the standard MLE analysis (Zhang, 2006) and its application for standard reward-based RL (Agarwal et al., 2020; Liu et al., 2024b). Recent works (Liu et al., 2024c; Xie et al., 2024; Cen et al., 2024) also applies this result for RLHF. For brevity, we omit the detailed proof here and direct readers to these related works for the proof. □

Combining (11), (12), (13), and Lemma C.3, together with the parameter choice $\alpha = 2/(\eta\exp(4R_{\max}))$, we obtain

$$
\mathcal{R}(T) \lesssim \frac{\beta T d_{\mathrm{PGEC}}}{\eta} + \beta\eta \cdot \exp(4R_{\max})\log(|\Pi|/\delta) + 4\sqrt{d_{\mathrm{PGEC}}T}
$$

$$
\lesssim \sqrt{d_{\mathrm{PGEC}} \cdot \exp(2R_{\max}) \cdot T \cdot \log(|\Pi|/\delta)},
$$

where the last line follows from the fact that $\eta = \sqrt{Td_{\mathrm{PGEC}}/(\exp(4R_{\max})\log(|\Pi|/\delta))}$. Therefore, we finish the proof of Theorem 5.3. □

# D  Experiment Setup

In experiments, we use the Alignment Handbook (Tunstall et al., 2023a) framework as our codebase. We find the best hyperparameter settings for the strong iterative DPO baseline by conducting a grid search

over the iteration number, batch size, learning rate, and label update rule. The results for the Zephyr-based models are shown in Figure 6. Specifically, we find that using the same amount of data, updating the model too many iterations can lead to instability. So we set the iteration number to 3 for Llama3-It-based and Zephyr-based models (excluding the first iteration of DPO training). Besides, we observe that choosing different batch sizes has a large effect on the models' performance and the optimal batch size heavily depends on the model architecture. In experiments, we set the batch size to 256 and 128 for the Zephyr-based and Llama3-It-based models, respectively. For the learning rate, we consider three design choices: cyclic learning rate with constant cycle amplitude, linearly decayed cycle amplitude, and decayed cycle amplitude at the last iteration. We find that a decaying cycle amplitude performs better than constant amplitudes in general. Thus, for Zephyr-based models, we set the learning to $5e-7$ for the first three iterations and $1e-7$ for the last iteration. In each iteration, the warmup ratio is 0.1. For Llama3-It-based models, we use a linearly decayed learning rate from $5e-7$ to $1e-7$ within 3 iterations with the same warmup ratio. We also test two update ways for the preference data. One is to rank $y_w, y_l, y_{\mathrm{ref}}$ and keep the best and worst responses in the updated dataset, which is the setting that is described in the main paper. The other is to compare $y_w$ and $y_{\mathrm{ref}}$ and replace the chosen or rejected response by $y_{\mathrm{ref}}$ based on the comparison result. We find that the former design performs better than the latter. We also compared with stepwise DPO (Kim et al., 2024), which updates the reference model at each iteration but uses the original dataset instead of the updated one. This demonstrates that exploring and collecting new data is necessary.

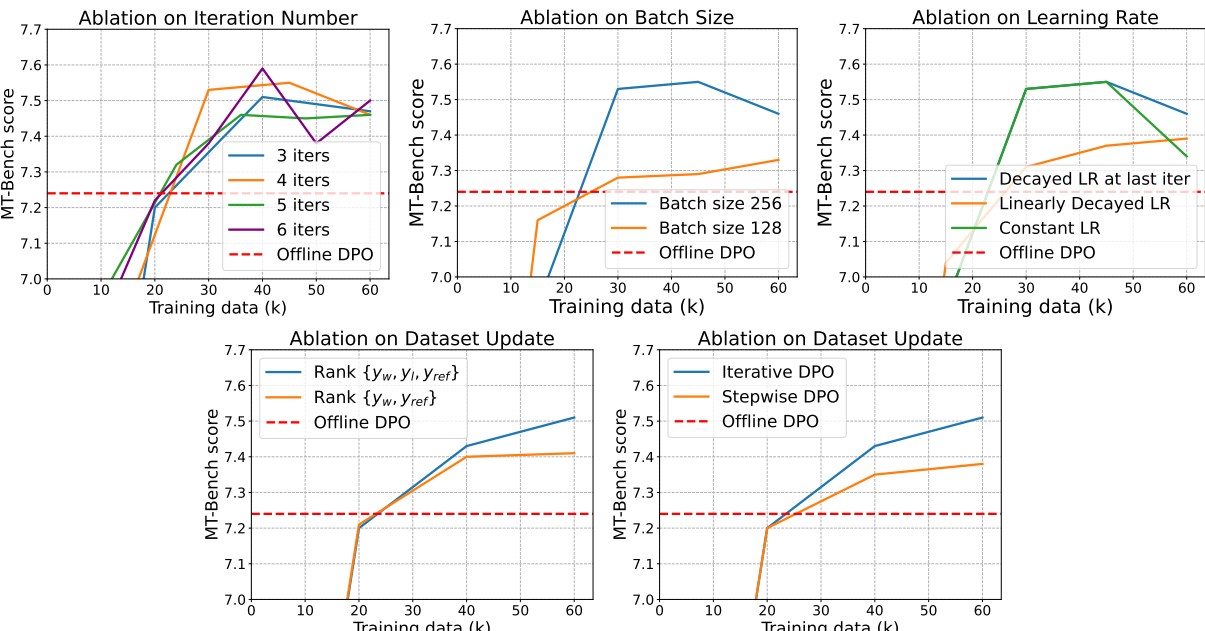

Figure 6: Ablation of the iterative DPO baseline. We conduct a grid search over the iteration number, batch size, learning rate, and designs of the dataset update rule.

For the proposed SELM method, we follow the above hyperparameter settings for a fair comparison. The optimism coefficient $\alpha$ is searched over 0.005, 0.001, 0.0005, and 0.0001 and is selected based on the average external reward on the holdout test set of UltraFeedback. We set $\alpha = 0.001$ for Zephyr-based SELM and $\alpha = 0.0001$ for Llama3-It-based SELM. For training SELM based on other models, we recommend setting $\alpha = 0.005$ or 0.001 as it shows minimal sensitivity to variations.

# E  Additional Experiments

## E.1  Ablation on Prompts Across Iterations

We conducted additional experiments on the data selection during online alignment. Specifically, for both the iterative DPO baseline and SELM, we update the policy on the same prompt dataset for multiple iterations. The results are shown in Table 3.

|  | MT-Bench |
|---|---|
| Zephyr-7B-SFT | 5.30 |
| Iterative DPO (same) | 7.21 |
| SELM (same) | 7.52 |
| Iterative DPO (diff) | 7.46 |
| SELM (diff) | 7.61 |

Table 3: MT-Bench average score when using the same set and different sets of prompts across iterations.

We find that the performance of both iterative DPO and SELM lags behind their counterparts that use varying prompts in different iterations. However, SELM is still able to outperform iterative DPO by a considerable margin.

## E.2  Ablation on Preference Pair Construction

Besides, we also implemented SELM following Algorithm 2 to sample six responses from the reference policy and use them both for calculating the optimism term in the objective and being ranked as preference pairs. The average scores on the MT-Bench are shown in Table 4.

|  | MT-Bench |
|---|---|
| Zephyr-7B-SFT | 5.30 |
| Iterative DPO | 7.46 |
| SELM (Alg. 1) | 7.61 |
| SELM (Alg. 2) | 7.55 |

Table 4: MT-Bench average score when using different strategies of preference pair construction.

It can be observed that both implementations of SELM demonstrate similar performance and consistently outperform iterative DPO. However, Algorithm 1, which we used in our main experiments, is more cost-efficient as it requires fewer samples to be drawn from the reference policy.

## E.3  Ablation on Optimistic Bias Versus Data Selection

We conducted additional ablations on the performance gains of SELM. Specifically, we study whether the gains come from the optimistically biased objective or the novel data as a result of exploration. We considered the following two settings: performing DPO on the preference data generated by the first iteration of SELM, and using the SELM objective on the original preference dataset. The results are shown in Table 5.

We observe that the data generated by SELM demonstrates superior performance when applied to both the DPO and SELM objectives, primarily due to its ability to encourage exploration. Preference optimization on SELM-generated data consistently outperforms optimization on data from the DPO model, suggesting that SELM's performance gains are driven more by enhanced exploration than by improvements in the

|  | MT-Bench |
|---|---|
| DPO Iter 1 | 7.53 |
| DPO Iter 2 | 7.55 |
| SELM Iter 1 | 7.53 |
| SELM Iter 2 | 7.61 |
| SELM Iter 1 data + DPO objective | 7.64 |
| DPO Iter 1 data + SELM objective | 7.51 |

Table 5: MT-Bench average score when using different objectives on data generated from different models.

optimization objective. In our main experiments, we use the SELM objective directly rather than applying DPO to SELM-generated data, as this approach avoids the need to train multiple models at each iteration.

Additionally, we measure the semantic similarity of the generated responses between SELM and the base model, as well as between DPO and the base model, using Sentence-BERT[3]. The results show that SELM achieves a similarity score of 0.69, which is lower than the 0.77 similarity score observed for DPO. This indicates that SELM facilitates more effective exploration compared to DPO.

---

[3]https://huggingface.co/sentence-transformers/all-MiniLM-L6-v2

