# OpenReview forum: "Self-Exploring Language Models: Active Preference Elicitation for Online Alignment"
_TMLR — Accepted by TMLR_

### Review · Reviewer_z5vv · 2024-12-21

**Summary Of Contributions:**

This paper introduces SELM, an LLM alignment framework that adds an optimistic bias term to DPO's objective. The authors claim this bias enables more effective exploration beyond DPO's passive sampling, leading to better-aligned models. The method is theoretically analyzed and evaluated on instruction-following benchmarks with Zephyr-7B and Llama-3-8B models.

**Audience:**

Yes

**Claims And Evidence:**

Yes

**Requested Changes:**

1. Bridge the theory-practice gap by either modifying the method to better match the theoretical analysis or revising the theory to explain the ranking mechanism.

2. Analyze generated responses qualitatively and quantitatively to demonstrate SELM enables more effective exploration than DPO.

3. Conduct ablations isolating gains from optimistic bias versus data selection to validate the core mechanism.

**Strengths And Weaknesses:**

Strengths

1. The theoretical analysis provides formal guarantees about sample efficiency, connecting exploration in preference optimization to established RL theory.

2. The method is practically implementable and integrates well with existing DPO framework, requiring no separate reward model.

3. Comprehensive ablation studies examine how the optimism coefficient α affects reward distributions and model behavior.

Weaknesses

1. While theory emphasizes exploration through optimistic bias, the actual algorithm operates more like a ranking-based data selection strategy.

2. Improvements over iterative DPO are marginal (e.g., +0.07 on MT-Bench for Llama-3-8B), and there's no evidence these gains come from better exploration rather than other factors.

3. Insufficient validation of core claims. No analysis demonstrates that SELM's exploration is more effective than DPO's inherent exploration through policy sampling.

---

> ### Author Response · Authors · 2025-01-12
> **Response to Reviewer z5vv**
>
> We thank the reviewer for identifying our work's novelty, soundness, and technical contributions. The valuable comments have helped us improve our manuscript (marked **blue** in the revision). Below are our specific responses to the questions raised by the reviewer:
>
> **Weakness 1: While theory emphasizes exploration through optimistic bias, the actual algorithm operates more like a ranking-based data selection strategy.**
>
> In the LLM alignment literature, which is the focus of this paper, reward models are commonly trained using human or AI feedback through preference optimization methods, such as the Bradley-Terry model (see Equations (2) and (3)), which rely on response rankings. Besides, one key contribution of our method lies in addressing the challenges posed by the inherent complexity and optimistic bias of bilevel optimization problems. Specifically, we reformulate this bilevel objective into a simpler and more tractable single-level optimization problem and reparameterize the reward model as a function of the policy. The theoretical results in Section 5 hold for both the original optimistically biased optimization objective and the derived single-level formulation.
>
> **Weakness 2: Improvements over iterative DPO are marginal (e.g., +0.07 on MT-Bench for Llama-3-8B), and there's no evidence these gains come from better exploration rather than other factors.**
>
> - The hyperparameters of iterative DPO are grid searched. This results in a strong baseline that matches SOTA online alignment algorithms fine-tuned from stronger base models. For instance, the MT-Bench score of `DPO-Iter-3 (Zephyr)` is $7.46$, which significantly improves `Zephyr-SFT` ($5.30$) by $2.16$, making it almost as good as DNO [1] ($7.48$). However, DNO is fine-tuned from the stronger model `Orca-2.5-SFT` ($6.88$), with an improvement of only $0.6$. Similarly, the MT-Bench score of SPPO [2] is $7.59$, which improves its base model `Mistral-7B-Instruct-v0.2` ($7.51$) by only $0.08$ after three iterations.
> - SELM consistently outperforms iterative DPO with minimal additional computing but higher sample efficiency. For example, even with increased data (3× and 1.5×, respectively), DPO-Iter-3 is outperformed by SELM-Iter-2 and SELM-Iter-1 on AlpacaEval:
>
> | SELM vs DPO Win Rate | DPO-Iter-3 | DPO-Iter-2 | DPO-Iter-1 | Zephyr-7B-DPO |
> |----------------------|------------|------------|------------|---------------|
> | SELM-Iter-2          | 53.26      | 53.59      | 58.33      | 65.56         |
> | SELM-Iter-1          | 52.32      | 49.65      | 53.91      | 64.43         |
> - To further address the reviewer's concerns, we conducted additional ablations to validate that the performance gains of SELM mainly come from better exploration. Specifically, we considered the following two settings: performing DPO on the preference data generated by the first iteration of SELM, and using the SELM objective on the original preference dataset. The results are shown in the following table:
>
>  |                      | MT-Bench|
> |----------------------|----------|
> | DPO Iter 1       | 7.53    |
> | DPO Iter 2       | 7.55    |
> | SELM Iter 1       | 7.53    |
> | SELM Iter 2        | 7.61     |
> | SELM Iter 1 data +  DPO objective        | 7.64     |
> | DPO Iter 1 data +  SELM objective        | 7.51     |
> - The data generated by SELM demonstrates superior performance when applied to both the DPO and SELM objectives, primarily due to its ability to encourage exploration. Preference optimization on SELM-generated data consistently outperforms optimization on data from the DPO model, suggesting that SELM's performance gains are driven more by enhanced exploration than by improvements in the optimization objective. In our main experiments, we use the SELM objective directly rather than applying DPO to SELM-generated data, as this approach avoids the need to train multiple models at each iteration.

---

> > ### Author Response · Authors · 2025-01-12
> > **Response to Reviewer z5vv (cont.)**
> >
> > **Weakness 3: Insufficient validation of core claims. No analysis demonstrates that SELM's exploration is more effective than DPO's inherent exploration through policy sampling.**
> >
> > - In **Section 5**, we analyzed the difference between the exploration mechanisms between SELM and DPO. Specifically, DPO passively explores and indiscriminately favors unseen responses. The exploration efficiency is thus low due to the vast space of natural language. In contrast, SELM actively explores by avoiding responses that have low implicit rewards.
> > - The above claim is also supported by experiments. We studied the difference in the implicit reward between SELM and DPO on the chosen and rejected response of the UltraFeedback test set in **Figure 4**. The results indicate that SELM
> > assigns a higher implicit reward than DPO for both responses. Besides, **Figure 5** shows that SELM consistently achieves higher implicit rewards across most prompts, with the positive reward differences being notably larger in magnitude, supporting that compared to DPO, SELM actively explores by prioritizing high-reward responses.
> >
> >
> > **Request Change 1: Bridge the theory-practice gap by either modifying the method to better match the theoretical analysis or revising the theory to explain the ranking mechanism.**
> >
> > Please see our response to **Weakness 1**. We thank the reviewer's suggestion and have added the above discussions to **Section 3-5**.
> >
> > **Request Change 2: Analyze generated responses qualitatively and quantitatively to demonstrate SELM enables more effective exploration than DPO.**
> >
> > - In **Figure 4** and **Figure 5**, we demonstrated that SELM assigns a higher implicit reward than DPO for responses in the test set and SELM consistently achieves higher implicit rewards across most prompts,  supporting that compared to DPO, SELM actively explores by prioritizing high-reward responses.
> > - Additionally, we measure the semantic similarity of the generated responses between SELM and the base model, as well as between DPO and the base model, using Sentence-BERT. The results show that SELM achieves a similarity score of $0.69$, which is lower than the $0.77$ similarity score observed for DPO. This indicates that SELM facilitates more effective exploration compared to DPO. The results are added to **Appendix E.3**.
> >
> > **Request Change 3: Conduct ablations isolating gains from optimistic bias versus data selection to validate the core mechanism.**
> >
> > Please see our response to **Weakness 2**. We thank the reviewer's suggestion and have added the additional experiments and discussions to **Appendix E.3**.
> >
> > ---
> > We hope the above responses and changes resolved the reviewer's concerns. We would be happy to have further discussions if the reviewer has any additional questions or comments.
> >
> > ---
> > [1] Rosset et al. "Direct Nash Optimization: Teaching Language Models to Self-Improve with General Preferences."\
> > [2] Wu et al. "Self-Play Preference Optimization for Language Model Alignment."

---

> > > ### Comment · Reviewer_z5vv · 2025-01-18
> > >
> > > After reviewing the rebuttal, I am now satisfied with the evidence provided and believe it adequately supports the claims in the paper.

---

### Review · Reviewer_PkQt · 2024-12-28

**Summary Of Contributions:**

In this paper, the authors study the setting of iterative online alignment for Large Language Models.
- They derive an objective to update the LLM policy that is reward-free (similar to DPO) and incorporates an additional optimism term to support the exploration in out-of-distribution regions with high implicit reward.
- With some modifications to the proposed bilevel optimization objective, they design a practical algorithm called SELM, which can be used for online preference alignment of modern LLMs.
- They provide an analysis of the gradient of their proposed objective and show how the exploration term in SELM addresses DPO’s known issue of decreasing the likelihood of responses generated by the reference policy.
- In the experimental section, they demonstrate that SELM outperforms other online alignment techniques, including iterative DPO.

**Audience:**

Yes

**Claims And Evidence:**

Yes

**Requested Changes:**

I would like to ask the authors to address and clarify the weaknesses and open questions/points.

**Strengths And Weaknesses:**

### Strengths:

- It is well-written and the motivation for improving exploration in online alignment is clearly presented.
- They provide a justification of why the exploration term helps based on the gradient analysis and based on a comparison with DPO.
- The experimental results are extensive. The authors use their algorithm to fine-tune two state-of-the-art base models and compare the performance on two instruction-following and other academic benchmarks.
- The authors include an ablation study on the optimism coefficient $\alpha$ to provide further insights into its impact on performance. Additionally, they conduct an ablation on the implicit reward $\hat{r}_{\theta}$ of SELM and DPO to provide insights into the learning dynamics.

### Weaknesses / Discussion Points:

- It seems that the main baseline for comparison is the custom iterative DPO method. I would have expected a more comprehensive comparison with other state-of-the-art online alignment baselines.
- Following the previous point, it would be helpful to include additional details explaining the baselines compared with their method, such as SPIN, DNO, and SPPO.
- To clarify, doesn’t the additional second term in the gradient in Equation $4.3$ become zero for high $\hat{r}_\theta$ due to exp$(-\hat{r}/\beta)$? Or is this controlled by the value of $\beta$? Have you observed the training to become more sensitive to the choice of $\beta$ compared to DPO?
- Could you explain the reasoning behind the selection of $3$ iterations for the online alignment in practice?

Minor comments:
- I recommend using global numbering for the equations.

---

> ### Author Response · Authors · 2025-01-12
> **Response to Reviewer PkQt**
>
> We thank the reviewer for identifying our work's novelty, soundness, and technical contributions. The valuable comments have helped us improve our manuscript (marked **blue** in the revision). Below are our specific responses to the questions raised by the reviewer:
>
> **Weakness 1:  It seems that the main baseline for comparison is the custom iterative DPO method. I would have expected a more comprehensive comparison with other state-of-the-art online alignment baselines.**
>
> - In addition to the strong iterative DPO baseline that adopts grid-searched hyperparameters, we also compared with various SOTA online alignment baselines in Table 1 and 2, including SPIN [1], DNO [2], and SPPO [3]. We copied some of the results in the following table:
>
> |                   | LC Win Rate | MT-Bench |
> |-------------------|-------------|----------|
> | Zephyr            | 8.01        | 5.30     |
> | iDPO (Zephyr)     | 22.19       | 7.46     |
> | **SELM (Zephyr)** | **24.25**   | **7.61** |
> | SPIN (Zephyr)     | 7.23        | 6.54     |
> | Orca              | 10.76       | 6.88     |
> | DNO (Orca)        | 22.59       | 7.48     |
> | Mistral           | 19.39       | 7.51     |
> | SPPO (Mistral)    | 28.53       | 7.59     |
>
> -   The results show that using `SELM` to fine-tune weaker models, such as `Zephyr`, can result in performance that surpasses online alignment algorithms fine-tuned from stronger models, such as `DNO` fine-tuned from `Orca-2.5` and `SPPO` fine-tuned from `Mistral-7B-Instruct-v0.2`. For example, the average MT-Bench score of `SELM (Zephyr)` is 7.61, which significantly improves `Zephyr` (5.30) by 2.31. In contrast, the scores of DNO and SPPO are 7.48 and 7.59, respectively, improving their stronger baselines (6.88 and 7.51) by 0.6 and 0.08.
>
> **Weakness 2: Following the previous point, it would be helpful to include additional details explaining the baselines compared with their method, such as SPIN, DNO, and SPPO.**
>
> Please see our response to **Weakness 1**. We thank the reviewer's suggestion and have added the above discussions to **Section 6.2**.
>
> **Weakness 3: To clarify, doesn’t the additional second term in the gradient in Equation (7)  become zero for high $\hat{r}_\theta$ due to $exp(−\hat{r}/\beta)$? Or is this controlled by the value of  $\beta$? Have you observed the training to become more sensitive to the choice of $\beta$ compared to DPO?**
>
> - The additional gradient term in Equation (7) is $-exp(−\hat{r}\_\theta(x, y)/\beta)$. When $\hat{r}\_\theta(x, y)$ is large, the gradient approaches zero, preventing $\log\pi_\theta(y\mid x)$ from decreasing. Conversely, as $\hat{r}_\theta(x, y)$ decreases, the policy reduces the likelihood of generating $y$, thereby prioritizing exploration of the high-quality uncertain responses.
> - This also aligns with our theoretical analysis in Section 5.1. Specifically, DPO indiscriminately favors unseen extrapolations and passively explores, while SELM performs active exploration by avoiding generating low-reward responses, improving the exploration efficiency.
> - In line with standard practice for DPO, we set $\beta$ to $0.01$. In our experiments, variations in $\beta$ did not reveal significant performance trends for either DPO or SELM, nor did SELM exhibit greater sensitivity to $\beta$ compared to DPO.
>
> **Weakness 4: Could you explain the reasoning behind the selection of $3$ iterations for the online alignment in practice?**
>
> In our ablation studies in **Appendix D**, we experimented with different numbers of iterations, including $3$, $4$, $5$, and $6$. The results indicated that $3$ iterations yielded the best overall performance for Iterative DPO. Consequently, for SELM, we also opted for $3$ iterations to maintain consistency and fair comparison.
>
> **Minor:  I recommend using global numbering for the equations.**
>
> Following the reviewer's suggestions, we have changed to global numbering.
>
> ---
> We hope the above responses and changes resolved the reviewer's concerns. We would be happy to have further discussions if the reviewer has any additional questions or comments.
>
> ---
> [1] Chen et al. ''Self-Play Fine-Tuning Converts Weak Language Models to Strong Language Models.''\
> [2] Rosset et al. "Direct Nash Optimization: Teaching Language Models to Self-Improve with General Preferences."\
> [3] Wu et al. "Self-Play Preference Optimization for Language Model Alignment."

---

### Review · Reviewer_3cNb · 2024-12-30

**Summary Of Contributions:**

In this paper, the authors propose a new bilevel optimization objective for alignment methods of LLMs. This is achieved by adding an optimism term to the reward-fitting loss. By doing similar derivation with DPO, the authors derived a RM-free objective which consistent of the orignial DPO objective together with an optimism term which is the negative log likelihood of the current policy on the pairs generated by the previous reference policy. This new objective does not bring much additional computational costs compared with naive DPO while it can allow the model to enjoy the benefits of active exploration. Empirical experiments show that the new derived objective can effectively improve the alignment performance compared with naive DPO. To summarize, the authors make the following contributions:

- The authors propose a new bilevel optimization objective which allows the LLM to perform active exploration during the alignment phase.
- The authors derive a simple loss for the proposed bilevel optimization problem which only requires to add an additional term that is easy to calculate compared with DPO.
- The authors conducted empirical experiments to show that the proposed methods can effectively achieve better performance compared with DPO.

**Audience:**

Yes

**Claims And Evidence:**

Yes

**Requested Changes:**

- Please add explanation of the new objective in 4.1, and explain the different of 4.1 and 1.1 and why 4.1 can still allow LLMs to perform active exploration.
- Please conduct experiments and analysis when the new response $y$ is from either the negative or the positive response of the reference policy.
- Please explain how the active exploration here is different from previous standard exploration techniques such as UCB in context bandit in the online settings.

**Strengths And Weaknesses:**

**Strength**
- The motivation of the proposed objective is very clear and the authors explained the motivation with the illustrated example intuitively.
- The whole derivation is quite clear and easy to follow.
- The experiment conducted in the paper is clear to follow and understand.

**Weakness**
- The objective in 4.1 is different from that of 1.1, especially change from the original reward objective to the relative objective, which I am not sure if the intuition still exists.
- For the additional term, in Algorithm 1, the authors directly sample another response $y$ from the reference policy. However, as we know both the negative or the positive pairs could be from the reference policy, so it is unclear what if we replace the new response with existing responses.

---

> ### Author Response · Authors · 2025-01-12
> **Response to Reviewer 3cNb**
>
> We thank the reviewer for identifying our work's novelty, soundness, and technical contributions. The valuable comments have helped us improve our manuscript (marked **blue** in the revision). Below are our specific responses to the questions raised by the reviewer:
>
> **Weakness 1: The objective in 4.1 is different from that of 1.1, especially change from the original reward objective to the relative objective, which I am not sure if the intuition still exists.**
>
> The optimistic reward objective presented in the introduction is used to help better understand our motivation and we make two practical modifications to it for our method:
> - Firstly, we incorporate a KL regularization term $\beta D_{\text{KL}}(\pi \,|| \,\pi_{\text{ref}})$ that is commonly used in RLHF to penalize large deviation between $\pi$ and $\pi_{\text{ref}}$.
> - Secondly, we replace the optimistic bias term $\max_\pi E_{y\sim\pi}[r(x, y)]$ with $\max_\pi E_{y\sim\pi, y'\sim\pi_{\text{ref}}}[r(x, y)-r(x, y')]$. The latter formulation effectively cancels out the reward signal for responses that are similar to $\pi_{\text{ref}}$. This relative reward term will only be strongly positive when $\pi$ discovers responses that get higher rewards than responses from the reference policy, thus creating a stronger incentive for exploration.
> - These modifications allow us to derive the closed-form inner solutions of the bilevel objective, enabling a reparameterization that yields the single-level objective presented in Equation (6), which is more practical for implementation.
> - Moreover, beyond the intuitive benefits, the modified objectives in Equations (5) and (6) are supported by theoretical guarantees. These include mitigating the indiscriminate preference for unseen responses while achieving provable high exploration efficiency.
>
> **Weakness 2: For the additional term, in Algorithm 1, the authors directly sample another response $y$  from the reference policy. However, as we know both the negative or the positive pairs could be from the reference policy, so it is unclear what if we replace the new response with existing responses.**
>
>  - Sampling responses from the reference policy and using them to both calculate the optimism term in the objective and as preference pairs align with our theoretical version of our framework in Algorithm 2.
>  - In the practical implementation Algorithm 1, we collect a single response from the reference policy and use it to calculate the optimism term and determine the preference pairs along with the original pairs from the dataset. We found that these two implementations have similar performance. We mainly conduct experiments following the setting in Algorithm 1 since it is less sensitive to sampling temperatures and more efficient.

---

> > ### Author Response · Authors · 2025-01-12
> > **Response to Reviewer 3cNb (cont.)**
> >
> > **Request Change 1: Please add explanation of the new objective in 4.1, and explain the different of 4.1 and 1.1 and why 4.1 can still allow LLMs to perform active exploration.**
> >
> > We added the following explanation and justification at the beginning of **Section 4.1**:
> > "To account for the change of $\pi$ relative to the reference policy $\pi_{\text{ref}}$, we introduce two modifications: (1) replacing the optimistic bias term $\max_\pi E_{y\sim\pi}[r(x, y)]$ with $\max_\pi E_{y\sim\pi, y'\sim\pi_{\text{ref}}}[r(x, y)-r(x, y')]$, and (2) incorporating a KL-divergence loss term between $\pi$ and $\pi_{\text{ref}}$ to minimize the deviation between $\pi$ and $\pi_{\text{ref}}$. Switching to relative rewards creates a stronger incentive for exploration, as they become significant only when $\pi$ discovers responses that outperform those from $\pi_{\text{ref}}$. Intuitively, it diverges from Equation (1) primarily in scenarios where the reward of responses from $\pi$ is high, but the relative reward is low. This signals that responses from $\pi_{\text{ref}}$ already receive high rewards and further exploration in this region is unnecessary."
> >
> >
> > **Request Change 2: Please conduct experiments and analysis when the new response $y$ is from either the negative or the positive response of the reference policy.**
> >
> > - Following the reviewer's suggestions, we implemented SELM that samples six responses from the reference policy and uses them both for calculating the optimism term in the objective and being ranked as preference pairs (i.e., Algorithm 2). The average scores on the MT-Bench are shown in the following table:
> >
> > |                      | MT-Bench |
> > |----------------------|----------|
> > | Zephyr-7B-SFT        | 5.30     |
> > | Iterative DPO | 7.46     |
> > | SELM (Alg. 1)          | 7.61     |
> > | SELM (Alg. 2)          | 7.55     |
> >
> > - It can be observed that implementations of SELM demonstrate similar performance and consistently outperform iterative DPO. However, Algorithm 1, which we used in our main experiments, is more cost-efficient as it requires fewer samples to be drawn from the reference policy.
> > - We have inforporated the above results in **Appendix E.2**.
> >
> > **Request Change 3: Please explain how the active exploration here is different from previous standard exploration techniques such as UCB in context bandit in the online settings.**
> >
> > We added the following explanation in the related work **Section 2**:
> > "UCB typically involves explicitly maintaining confidence sets that contain the ground truth with high probability, which poses considerable challenges for implementation especially in high-dimensional spaces such as natural language. In contrast, our method only optimizes a single objective that integrates reward estimation and planning to automatically balance exploration and exploitation."
> >
> > ---
> > We hope the above responses and changes resolved the reviewer's concerns. We would be happy to have further discussions if the reviewer has any additional questions or comments.

---

### Review · Reviewer_aVa1 · 2025-01-07

**Summary Of Contributions:**

This paper study the problem of exploration in online preference alignment of large language models (LLMs). The authors propose Self-Exploring Language Models (SELM), which minimizes the original DPO loss while reducing the probability of generated answers using the current policy (Eq. (4.2)) to achieve exploration.

The authors show that (Theorem 5.3) SELM can be guaranteed to achieve sublinear cumulative regret under Assumption 5.2.

The authors conducted experiments on (LLM-as-a-judge tasks such as AlpacaEval 2.0, and MT-Bench and several other standard tasks (Table 2), using Llama-3-8B-Instruct and Zephyr-7B-SFT as the base models. The results show that SELM gains further improvements over DPO.

**Audience:**

Yes

**Broader Impact Concerns:**

I do not have concerns on the ethical implications of the work.

**Claims And Evidence:**

Yes

**Requested Changes:**

Algorithm 1 contains "for iteration t = 1, 2, ..., T" which seems to indicate there are many iterations. The introduction also gives audience this impression, since exploration is a problem where one context is encountered for many times and the policy has multiple chance to explore and then eventually figure out a good answer.

However, as also noted in the paper, the prompt dataset was split into three equal-size parts, and then SELM only went through the dataset once, and T = 3 (there are only 3 iterations in Algorithm 1). This doesn't seem to be an "online alignment", since in this case each context will only be encountered once, and after that the policy will never have another chance to explore another answer?

Maybe a better way to align the experiments and theory study is to make the implementation online, such that the policy can go over and over the prompt dataset for multiple times, which will be the true version of online alignment.

**Strengths And Weaknesses:**

**Strengths**

Exploration in online preference alignment is a topic worth studying. This work makes a reasonable contribution to this topic.

The experimental results look good, which verify the proposed method.

The paper is clearly written and easy to follow.

**Weaknesses**

The implementation and experiments seem not the same as the online setting studied.

---

> ### Author Response · Authors · 2025-01-12
> **Response to Reviewer aVa1**
>
> We thank the reviewer for identifying our work's novelty, soundness, and technical contributions. The valuable comments have helped us improve our manuscript (marked **blue** in the revision). Below are our specific responses to the questions raised by the reviewer:
>
> **Weakness 1: The implementation and experiments seem not the same as the online setting studied.**
>
> In this paper, the online setting specifically refers to iterative or batched online RLHF, as established in previous works [1, 2, 3]. We opted against adopting purely online settings due to efficiency concerns. Purely online approaches require generating samples at every step of the updated LLM, diminishing the efficiency benefits offered by batched inference.
>
> **Request Change 1: Algorithm 1 contains "for iteration t = 1, 2, ..., T" which seems to indicate there are many iterations. The introduction also gives audience this impression, since exploration is a problem where one context is encountered for many times and the policy has multiple chance to explore and then eventually figure out a good answer. However, as also noted in the paper, the prompt dataset was split into three equal-size parts, and then SELM only went through the dataset once, and T = 3 (there are only 3 iterations in Algorithm 1). This doesn't seem to be an "online alignment", since in this case each context will only be encountered once, and after that the policy will never have another chance to explore another answer? Maybe a better way to align the experiments and theory study is to make the implementation online, such that the policy can go over and over the prompt dataset for multiple times, which will be the true version of online alignment.**
>
> - Our choice of using different subsets during online alignment follows previous works that iteratively align the LLM [1, 2, 3]. Reusing the same data across iterations increases the risk of overfitting the model to that specific data distribution. By introducing new data, the algorithm ensures exposure to diverse samples, improving generalization and robustness.
> - Following the reviewer's suggestions, we conducted additional ablations on the data selection strategy during online alignment. Specifically, for both the iterative DPO baseline and SELM, we update the policy on the same prompt dataset for multiple iterations (denoted as "same" in the table). The average scores on the MT-Bench are shown in the following table:
>
> |                      | MT-Bench |
> |----------------------|----------|
> | Zephyr-7B-SFT        | 5.30     |
> | Iterative DPO (same) | 7.21     |
> | SELM (same)          | 7.52     |
> | Iterative DPO (diff) | 7.46     |
> | SELM (diff)          | 7.61     |
> - It can be observed that the performance of both iterative DPO and SELM lags behind their counterparts that use varying prompts in different iterations. However, SELM is still able to outperform iterative DPO by a considerable margin.
> - We thank the reviewer for the suggestions and have added the above experiments and discussions to Appendix E.1.
>
> ---
> We hope the above responses and changes resolved the reviewer's concerns. We would be happy to have further discussions if the reviewer has any additional questions or comments.
>
> ---
> [1] Bai et al. ''Training a helpful and harmless assistant with reinforcement learning from human feedback.''\
> [2] Gulcehre et al. "Reinforced self-training (rest) for language modeling."\
> [3] Xiong et al. "Gibbs sampling from human feedback: A provable kl-constrained framework for rlhf."

---

### Comment · Action_Editor_vy8k · 2025-01-14
**Please take a look at the rebuttal**

Dear reviewers,

The authors have submitted their responses as well as a revised version of the paper. Please take a look at them and let the authors know if you still have any questions or you agree/disagree with them. The goal is to minimize any uncertainty you have about the paper, so you can make a better recommendation.

Thank you for your help in reviewing this paper!

Amir-massoud (Action Editor)

---

### Decision · Action_Editor_vy8k · 2025-02-25

**Recommendation:** Accept with minor revision

**Comment:**

**Brief Summary:**

The paper proposes an LLM alignment procedure through preference feedback in the online setting when more data can be collected. The online setting brings the question of efficient exploration. Self-Exploring Language Model (SELM) is a Reward Model-free online alignment method equipped with an active exploration mechanism. The paper theoretically studies a variant of SELM and empirically studies its performance.

**Summary of reviewer recommendations:**

We have four recommendations: 1x Accept, 2x Leaning Accept, and 1x Leaning Reject.
The positive reviews mentioned that they were satisfied with the authors rebuttal and the revisions in the paper.
The negative review mentions that "I am not convinced by the response that 1) the motivation of change the objective which is different from before; it looks like the changing is mainly due to derivation 2) the authors did not conduct the experiment as I requested (use either of the old sample instead of sampling new ones.)".

As we did not have a consensus, I read the paper myself.

I believe the clarification regarding the motivation of the change of objective is satisfying enough. I can also see that the authors added new experiments to address the negative reviewer's concerns (Appendix E.2). Whether this is exactly what the reviewer hoped for or not is not completely clear as the reviewer did not have a discussion with the authors.

In my own reading of the paper, I also agree that the paper has a novel contribution and a good execution overall. But I also found a few points that require further clarifications before the paper can be published. Several of them are minor.


- Algorithm 1 allows any online iterations T. As far as I understand, the experiments use T = 3. This should be noted much earlier in the paper, perhaps even in the last paragraph of Introduction when the experiments are summarized.

- The regret is defined w.r.t. the optimal policy $\pi^*$ (Section 5.2). Is this the optimal policy of regularized objective (Eq. 4) or w.r.t. $\beta = 0$? I have the same question for $\pi^*$ appearing in the proof of Theorem 5.3 in Appendix C.1.

- Theorem 5.3 has the size of policy class $\Pi$. Nowhere it is assumed or mentioned that the policy class is finite. I understand that this finiteness assumption simplified the result, but if it is required for the results to be non-vacuous, it should be explicitly mentioned too.
Later, as an example, a log-linear model is introduced. For that class, the size is infinite. A clarification is needed.

- In Appendix C – Proof of Theorem 5.3, it is mentioned that "We emphasize that it is not a novel contribution of the present work". What is not a novel contribution? The "proof technique" or Theorem 5.3 or the analysis of Algorithm 2?

- The difference with Xie et al. (2024) needs more discussions. For example, it is mentioned that "[T]he theoretical version of the self-exploration mechanism (Algorithm 2) is a bit different from the practical one used in the numerical experiments and is closer to the proposed algorithm in Xie et al. (2024)".
This makes me wonder what the similarities and differences with Xie et al.'s XPO algorithm are.
Please carefully expand on this, delineate similarities and differences of both the algorithms and their theoretical results.


- Why does equality (10) hold? This seems to be Lemma C.3 of Xie et al. (2024) or at least a variant of it. If it is, please cite it. If not, please prove it.

- It is not clear to me how Eq. (10) implies the equality beforehand. Please clarify it.

- What is $Z_r$ in Eq. (10)?

- What is the meaning of an informal theorem (Theorem C.2)? Has this been proved? If not, it is a conjecture, and should be stated as such.

- During the rebuttal phase, some additional experiments were added in Appendix E. As far as I can see, there is no pointer from the main text to the results of this appendix. Please help the reader appreciate your results by briefly describing them much earlier in the paper.

Minor:
- Notations such as x and y are not defined when used in Eq. (1). Likewise is y_u on page 2.
- In Algorithm 2, Line 3, it is not clear which of $y_t^1$ and $y_t^2$ is the accepted and rejected response.

- Typo: In the statement of Theorem C.2, we have "toekn-wise".
- Typo: On page 21, we have "handsise".


Overall, I recommend acceptance of the paper after these clarifications have been made.

**Audience:**

Yes. This paper would be interesting for the readers who are interested in LLMs and preference alignment.

**Claims And Evidence:**

Mostly, though there are some points regarding the contributions of the paper where more clarifications are needed.

---

> ### Author Response · Authors · 2025-03-03
> **Clarifications and Revisions for the Camera-Ready Version**
>
> We thank the Action Editor for the time and valuable feedback. We have uploaded the camera-ready version of our manuscript, which contains the following clarifications and revisions:
>
> **Point 1: Algorithm 1 allows any online iterations T. As far as I understand, the experiments use T = 3. This should be noted much earlier in the paper, perhaps even in the last paragraph of Introduction when the experiments are summarized.**
>
> We have clarified our experimental settings by explicitly stating in the final paragraph of the Introduction that three iterations are performed. In addition, Section 4.2 now emphasizes that Algorithm 1 is instantiated with $T=3$ in our experiments.
>
> **Point 2: The regret is defined w.r.t. the optimal policy (Section 5.2). Is this the optimal policy of regularized objective (Eq. 4) or w.r.t. $\beta = 0$? I have the same question for the policy appearing in the proof of Theorem 5.3 in Appendix C.1.**
>
> $\pi^*$ is the optimal policy with respect to the regularized objective (Eq. 4), both in the main paper and the proof provided in the appendix. This clarification has been added to Section 5.2 where $\pi^*$ is first introduced.
>
> **Point 3: Theorem 5.3 has the size of policy class $\Pi$. Nowhere it is assumed or mentioned that the policy class is finite. I understand that this finiteness assumption simplified the result, but if it is required for the results to be non-vacuous, it should be explicitly mentioned too. Later, as an example, a log-linear model is introduced. For that class, the size is infinite. A clarification is needed.**
>
> For simplicity, our analysis assumes that the policy class is finite. We note, however, that this assumption can be relaxed to accommodate infinite classes by introducing covering number arguments. For instance, when considering a log-linear policy class, the effective complexity scales approximately with the dimension $d$ (see, e.g., [1]). This discussion has been incorporated into Assumption 5.2.
>
> **Point 4: In Appendix C – Proof of Theorem 5.3, it is mentioned that "We emphasize that it is not a novel contribution of the present work". What is not a novel contribution? The "proof technique" or Theorem 5.3 or the analysis of Algorithm 2?**
>
> Please refer to our response to Point 5 below.
>
> **Point 5: The difference with Xie et al. (2024) needs more discussions. For example, it is mentioned that "The theoretical version of the self-exploration mechanism (Algorithm 2) is a bit different from the practical one used in the numerical experiments and is closer to the proposed algorithm in Xie et al. (2024)". This makes me wonder what the similarities and differences with Xie et al.'s XPO algorithm are. Please carefully expand on this, delineate similarities and differences of both the algorithms and their theoretical results.**
>
> The main difference between our regret analysis and that of Xie et al., 2024 [2] lies in the complexity measure we used, which is known as GEC (Generalization Eluder Coefficient). On the other hand, the proof technique is a straightforward combination of the reduction technique in Xie et al. and the regret analysis in the general setting of reinforcement learning, which is detailed below. Hence, we do not emphasize it as our primary contribution.
>
> In the first version of our paper on arXiv, we did not have the regret analysis, which is a corollary of the regret analysis developed in a previous paper (Maximize to Explore: One Objective Function Fusing Estimation, Planning, and Exploration [3]). The previous paper focused on the general setting of reinforcement learning, rather than reinforcement learning from human feedback. Shortly after the first version of our paper in arXiv, Xie et al. was uploaded to arXiv with a regret analysis. In the second version of our paper submitted to TMLR, we added the regret analysis. In particular, we used the reduction approach developed in Xie et al., which reduced reinforcement learning from human feedback to reinforcement learning and bridged the regret analysis developed in the previous paper [3].

---

> > ### Author Response · Authors · 2025-03-03
> > **Clarifications and Revisions for the Camera-Ready Version (cont.)**
> >
> > **Point 6-8: Why does equality (10) hold? This seems to be Lemma C.3 of Xie et al. (2024) or at least a variant of it. If it is, please cite it. If not, please prove it. It is not clear to me how Eq. (10) implies the equality beforehand. Please clarify it. What is $Z_r$ in Eq. (10)?**
> >
> > Equation (10) is derived as the optimal solution of the KL-constrained reward maximization objective (Equation (4)), which we restate here for clarity:
> > $$\pi^*(y\mid x) := argmax_\pi E_{x\sim\mathcal{D}, y\sim\pi(\cdot|x)}[y\mid x] - \beta D_{\text{KL}}(\pi || \pi_{\text{ref}}) = \frac{1}{Z_r(x)}\pi_{\text{ref}}(y\mid x)\exp(r(x, y) / \beta),$$
> > where $Z_r(x) = \sum_y\pi_{\text{ref}}(y| x)\exp(r(x, y)/\beta)$ (as defined in Section 4.1). The proof follows standard arguments that appear in various previous works such as [4, 5] (e.g., Appendix A.1 in [4]). By applying a logarithmic transformation to both sides and rearranging terms, we arrive at Eq. (10). We have added the above clarification to Appendix C.1.
> >
> > **Point 9: What is the meaning of an informal theorem (Theorem C.2)? Has this been proved? If not, it is a conjecture, and should be stated as such.**
> >
> > Theorem C.2 is a conjecture, whose formulation is meant to capture a token-level version of the result presented in Theorem 5.3. But the proof technique for it is very similar to that of Theorem 5.3.
> >
> > **Point 10: During the rebuttal phase, some additional experiments were added in Appendix E. As far as I can see, there is no pointer from the main text to the results of this appendix. Please help the reader appreciate your results by briefly describing them much earlier in the paper.**
> >
> > We appreciate the suggestion to better integrate our additional experimental results. Accordingly, we have included a brief description and pointers to the ablation studies in Appendix E at the end of Section 6.
> >
> > ---
> > Once again, we sincerely appreciate your continued dedication and insightful feedback throughout the review and submission process, which has greatly improved our work.
> >
> > ---
> >
> > [1] Zanette et al. ''Provable Benefits of Actor-Critic Methods for Offline Reinforcement Learning.''\
> > [2] Xie et al. ''Exploratory Preference Optimization: Harnessing Implicit Q*-Approximation for Sample-Efficient RLHF.''\
> > [3] Liu et al. ''Maximize to Explore: One Objective Function Fusing Estimation, Planning, and Exploration.''\
> > [4] Rafailov et al. ''Direct Preference Optimization: Your Language Model is Secretly a Reward Model.''\
> > [5] Peng et al. ''Advantage-weighted regression: Simple and scalable off-policy reinforcement learning.''

---

> > > ### Comment · Action_Editor_vy8k · 2025-03-04
> > >
> > > Thank you! I appreciate your response to my concerns. I would be happy to accept the paper.
> > > I noticed that a few typos are still. If you do a round of proof-reading and submit a revised version of the paper soon, I will accept that version, so that the camera ready version is typo-free.
> > >
> > > Some examples that I spotted:
> > >
> > > - Typo: On page 1, we have "defacto" instead of "de facto".
> > > - Typo: On page 20, in the statement of Theorem C.2, we have "toekn-wise" instead of "token-wise".
> > > - Typo: On page 21, just after (13), we have "handsise" instead of "handside".

---

> > > > ### Author Response · Authors · 2025-03-05
> > > > **Typo fixed in camera ready revision**
> > > >
> > > > Dear Action Editor,
> > > >
> > > > Thank you for your time and valuable feedback! We have completed another round of proofreading and have corrected the typos, including the ones that you pointed out. We appreciate your careful review and have uploaded the typo-free camera-ready revision.
> > > >
> > > > Best regards,\
> > > > Authors of Submission 3629

---

> > > > > ### Comment · Action_Editor_vy8k · 2025-03-05
> > > > >
> > > > > Thank you!